# A bioabsorbable body-coupling-electrotherapy suture

Zhouquan Sun[1,3], Yuefan Jin[2,3], Hui Su[1,3], Yaogang Li[1], Qinghong Zhang[1], Kerui Li[1], Hongzhi Wang[1], Linpeng Li[2] ✉, Shan-kai Yin[2], Chengyi Hou[1,2] ✉ & Hui Wang[2] ✉

Sutures are pivotal medical devices for postoperative incision management. The inherent capacity of sutures to regulate wound healing and promote regeneration has the potential to significantly reduce patient discomfort and conserve medical resources. Here, we developed a bioabsorbable body-coupling electrotherapy suture that modulates healing from inflammation to remodeling, enabling healthy repair. The suture combines high tensile strength, flexibility, and degradability. The conversion of body-coupled electromagnetic energy through the suture enabled electrically synergistic therapeutic capabilities. By converting body-coupled electromagnetic energy, it provides synergistic therapy: dielectric polarization accelerates antimicrobial and anti-inflammatory effects, while dielectric voltage difference enhances healing factor expression. This fully physical approach reduces reliance on silver nanoparticles. In vitro and in vivo studies confirmed stable performance, achieving more than 1.43-fold improvement in healing efficiency compared with single-function sutures and reducing postoperative infections. Offering immobilization, stage-wide modulation, sustainability, and low energy demand, this technology represents a significant advancement in surgical practice.

As a necessary medical device in therapeutic domain, suture plays a critical role in wound closure and healing[1]. Its application is versatile, encompassing a wide spectrum of wound types, including cuts, lacerations, deep puncture wounds and postoperative wounds, as well as stitches from skin to deeper tissue, making it the most wide-used treatment compared to dressing therapy[2]. The process of successful wound healing following stitching is dependent on the coordinated actions of immunomodulation, cell migration and tissue regeneration[3,4]. But the majority of clinical sutures are designed solely for mechanical wound closure, lacking additional features to manage the healing process[5]. The development of sutures with intrinsic healing-promoting properties has the potential to significantly enhance the

field of intelligent healthcare. However, the development of functional sutures has lagged behind other smart therapeutic devices, such as functionalized gels[6–9], electronic patches[10–13] or bandages[14–17]. These have made significant breakthroughs in the field of wound healing by integrating multiple therapeutic functions (sustained drug/cytokine release, electrical stimulation, etc.) to regulate the healing process and achieve the desired outcome. In comparison, sutures have smaller dimensions (micron scale). A major challenge lies in the integration of multifunctionality into sutures while maintaining their stitchability and mechanical properties for such functional carriers[18].

Nevertheless, sutures have evolved from chemical antimicrobial sutures to physical electrical stimulation (ES) sutures[19]. The utilization

[1]State Key Laboratory of Advanced Fiber Materials, College of Materials Science and Engineering, Donghua University, Shanghai, PR China. [2]Department of Otolaryngology-Head and Neck Surgery, Otolaryngology Institute of Shanghai JiaoTong University, Shanghai Sixth People's Hospital Affiliated to Shanghai Jiao Tong University School of Medicine, Shanghai, PR China. [3]These authors contributed equally: Zhouquan Sun, Yuefan Jin, Hui Su. ✉e-mail: lilp@sjtu.edu.cn; hcy@dhu.edu.cn; wangh2005@sjtu.edu.cn

of chemical antimicrobial sutures, encompassing antibiotics[20], metal antimicrobials[21], photodynamic therapy (PDT)[22,23], has been widely reported as a means to modulate immune responses. However, persistent chemical hazards, such as tissue drug resistance, cytotoxicity of excessive metal nanoparticles and PDT molecules, are hindering the development of effective antimicrobial sutures. In response to these challenges, the self-powered ES sutures developed by our group[19] to modulate cell proliferation and migration have overcome the chemical hazards associated with antimicrobial sutures and the need for an external power source. However, since the mechanoelectric system relied on external mechanical stimulation to generate energy, it could only be applied in dynamic or specific activation scenarios. It is crucial to note that these pro-healing functions have been achieved independently due to the inherent differences in the mechanisms of action of existing antimicrobials and ES, and have not been able to achieve the same comprehensive wound management as other intelligent therapeutic devices. The next-generation functional sutures must be infused with another physiotherapeutic mechanisms to achieve clinical-grade suture characteristics. The bright side is that our group has proposed a form of energy interaction of body-coupled fiber[24]. Inexhaustible electromagnetic (EM) energy could be harvested and transmitted wirelessly through the fiber-body interface. Exploration of another form of in vivo energy conversion on this basis holds promise in addressing the aforementioned challenges.

Here, we developed a bioabsorbable body-coupling-electrotherapy suture (BET-suture) that has the capacity to wirelessly convert body-coupled EM energy into electrotherapy fields at the wound site for postoperative wound repair. We discussed the development of the body-coupled system in Supplementary Note S1. The suture consists of twisted absorbable core-sheath fibers, ensuring both flexibility (2.84 cN•cm$^2$/cm) and strength (1.52 GPa). We have enhanced the dielectric properties of the BET-suture using safe amounts of silver nanoparticles (Ag NPs), maximizing its ES and antimicrobial capabilities. Following implantation, the BET-suture undergoes self-adaptive dynamic polarization under body-coupled energy condition, generating a dielectric voltage difference to apply effective ES (0.75−5 V/mm) and promote tissue regeneration. Concurrently, the sheath's stored charges (5.07 μC/cm$^2$) following polarization could be utilized for antibacterial action and to induce conformational shifts in macrophages, thereby modulating the immune response, preventing wound infection and ensuring healthy tissue remodeling. This physical antimicrobial strategy addresses the potential biological hazards associated with the overuse of Ag NPs as antimicrobial agents. Furthermore, the clinical efficacy and molecular mechanism of action of BET-suture have been validated through cellular and animal studies. In short, BET-suture enables comprehensive physical therapy of wounds from the inflammatory phase to the remodeling phase through body-coupled energy in diverse conditions, providing a effective solution for the creation of intelligent surgical sutures.

## Results

### Design and principle of bioabsorbable body-coupling-electrotherapy suture

In daily life, a wide variety of electronic devices and energy transmission lines were ubiquitous, and they emitted EM fields to the surrounding environment when they were in operation. The human body, as a substantial dielectric, possessed the capacity to couple the EM energy from the environment and convert it into electrical output, which was sufficient to light up LEDs (Supplementary Fig. S1, Supplementary Note S1, Supplementary Movie S1). Even under complex E-fields, maintaining a safe distance would ensure personal safety (Supplementary Fig. S2a, b). Meanwhile, within the frequency range of biological electrical signals (10−90 Hz), it is possible to couple a stable voltage (Supplementary Fig. S2c). The coupled voltage is also

unaffected by the individual's movement state (Supplementary Fig. S2d, Supplementary Movie S2). It is envisaged that the implementation of such portable energy sources for ES of postoperative trauma would further save healthcare resources and improve treatment. In this regard, we developed an EM energy-driven absorbable wireless electrotherapy suture based on body coupling. BET-suture was composed of twisted core-sheath fibers with a conductive core (molybdenum, Mo) and a dielectric sheath (polylactic acid-glycolic acid/silver nanoparticles, PLGA/Ag NPs). The suture was able to be polarized by body-coupled ambient EM energy to store charges and generate a dielectric voltage difference to convert it into ES, addressing various challenges in wound treatment: (1) preventing bacterial infection; (2) accelerating cell migration and proliferation; and (3) promoting angiogenesis. Upon completion of treatment, BET-suture would be degraded and absorbed in the body without the need for a second surgical removal (Fig. 1a). Compared to previous antimicrobial sutures and self-powered ES sutures, the BET-suture introduces a comprehensive electrotherapy mode for wound healing management, adaptable to various therapeutic scenarios under both static and dynamic conditions, making it better aligned with clinical practice (Fig. 1e).

As a highly consumable medical material, BET-suture could be prepared in a continuous process. Primary fiber (Mo@PLGA/Ag NPs) was first obtained by attaching PLGA/Ag NPs composite to the surface of Mo filament using a modified wet-spinning technique to form a dielectric sheath layer with a thickness of about 15 μm. In order to apply this fiber to the stitching of desired sites, the primary fibers were further twisted to obtain secondary fibers (BET-suture) (Supplementary Fig. S3a–c). The corresponding physical photographs of BET-suture, primary and secondary fiber, cross-sectional SEM images and elemental distributions are shown in Fig. 1b, c and Supplementary Fig. S3d, respectively. The mean diameter of the primary fiber and the BET-suture with 5 strands was measured to be 65 μm and 201 μm, respectively.

BET-suture used for wound stitching must have high strength and adequate suturing properties. The electrode, serving as the core layer, plays a decisive role on the strength of BET-suture. Compared to BET-suture prepared using other biodegradable metals (magnesium, Mg and tungsten, W) as electrodes, BET-suture prepared using Mo filament as the electrode exhibited superior tensile strength (Supplementary Fig. S4a). Similarly, BET-suture showed higher mechanical strength compared to current types of medical sutures, with a breaking tensile force and strength of 21.4 N and 1.52 GPa, respectively (Supplementary Fig. S4b, c). As the number of strands increased, BET-suture showed higher strength to accommodate different requirements of suture sites (Supplementary Fig. S4d). Meanwhile, the breaking strength of BET-suture remained high (1.32 MPa) after knotting (Supplementary Fig. S4e), avoiding the situation where the suture was prone to break after stitching. Moreover, we measured the tissue drag force per unit circumference when the suture was pulled through different tissue models (Supplementary Fig. S4f). BET-suture showed a traction force range of 3.91–4.66 N/cm, which is in the range between that of medical nylon sutures (5.68 N/cm) and silk sutures (2.89 N/cm), and is in line with the requirements for medical sutures. We further measured the bending stiffness of the sutures using the Kawabata Evaluation System (Supplementary Fig. S5). BET-suture exhibited a bending stiffness value of 2.84 cN•cm$^2$/cm, which is close to the feel of commercial sutures and ensures flexibility (Fig. 1d). Meanwhile, its tensile stiffness is comparable to the reported state-of-the-art high strength and functionalized sutures (Supplementary Table S1).

We validated the biocompatibility and bioabsorbability of BET-suture. Supplementary Fig. S6a illustrated the degradation and metabolic mechanisms mechanism of BET-suture components. It was

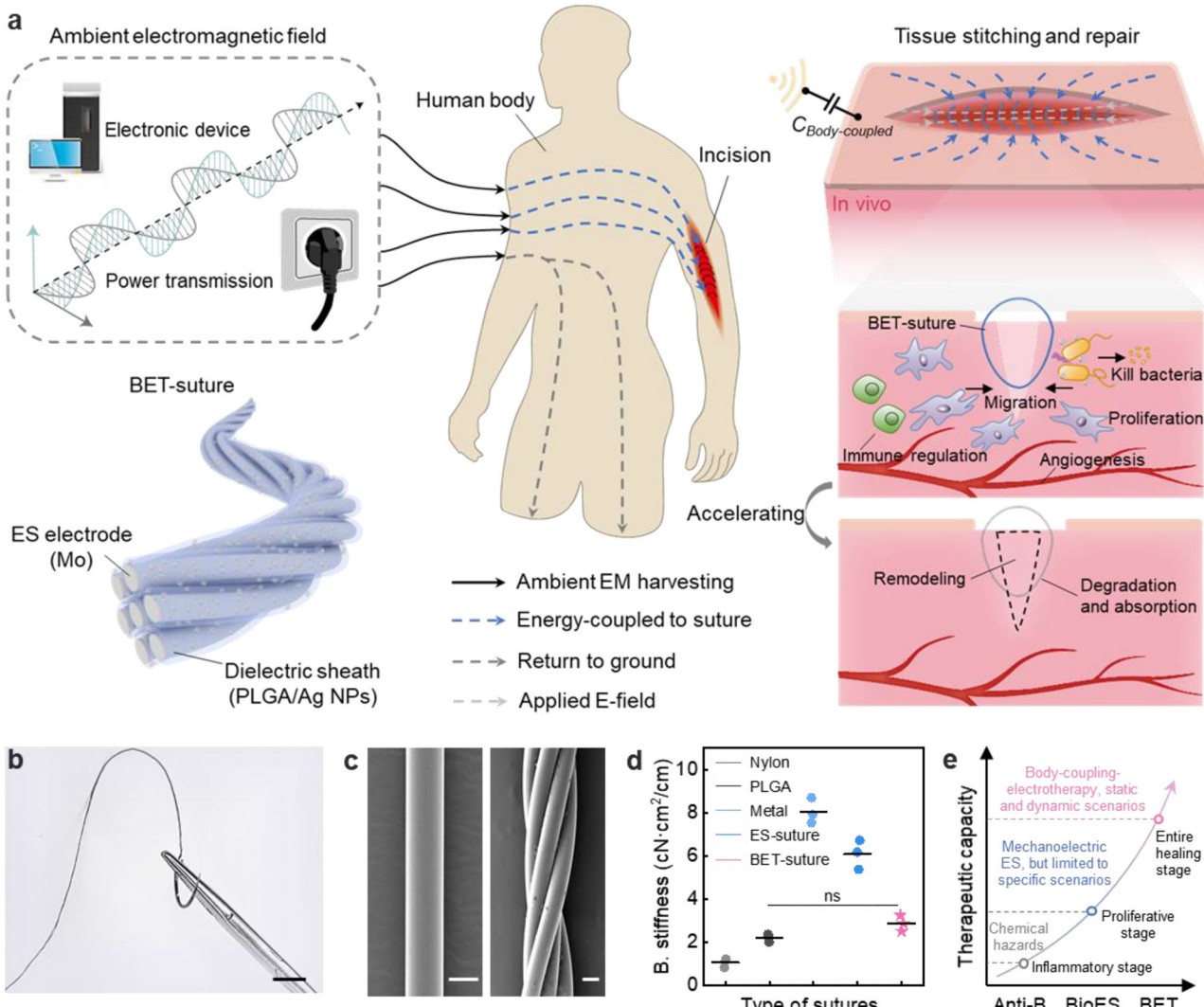

**Fig. 1 | Design and principle of bioabsorbable body-coupling-electrotherapy suture. a** BET-suture promoting tissue repair processes. BET-suture senses and stores EM energy from the body-coupled environment, generating an E-field at the wound site to perform reparative functions. The main effects: accelerating fibroblast migration and proliferation, inhibiting bacterial growth; promoting angiogenesis, down-regulating immune cell activity, and improving tissue remodeling. **b** Optical image of BET-suture. Scale bar: 2 cm. **c** SEM images of primary fiber (left) and BET-suture (right). Scale bar: 60 μm. $n = 3$ independent samples. **d** Bending stiffness of different types of sutures. $n = 3$ independent samples. **e** The development of suture technology−from antimicrobial chemical suture (Anti-B) technology to self-powered ES (BioES) suture to the currently used body-coupling-electrotherapy (BET) suture. All statistical analyzes were performed by one-way ANOVA, data represent mean ± standard deviation, ns indicates not significant.

noteworthy that excess Ag NPs may be cytotoxic in previous reports[25]. We measured the relationship between the amount of Ag NPs released from BET-suture and the degradation time (Supplementary Fig. S6b). As degradation process progressed, the amount of Ag NPs released ranged from 0 to 100 ppb, well below the reported safe levels (<1 ppm)[26]. This is due to the fact that BET-suture does not regulate the immune process through the release of Ag NPs, and Ag NPs were added at very low levels, as discussed in detail in part 2.3. After 28 days of degradation, the release of Ag NPs reached a maximum corresponding to almost complete hydrolysis of the sheath layer (Supplementary Fig. S6c). After 77 days, no intact BET-suture was observed in the phosphate-buffered saline (PBS) solution. We performed live/dead staining of degradation solution-treated cells with different contents of Ag NPs and compared them with BET-suture components. The results showed no statistically significant difference between the groups, demonstrating the biosafety of BET-suture and its degradation products (Supplementary Fig. S7).

## Mechanism of electrical stimulation based on dielectric voltage difference

Following the fulfilment of the fundamental prerequisites for medical sutures, we analyzed the electrotherapeutic mechanism of BET-suture. The normal effect of ES requires the establishment of a potential difference between the wound and the surrounding tissues to constitute an endogenous E-field[27]. After implantation of BET-suture, body-coupled EM energy would be transmitted into the electrodes of BET-suture. The energy storage properties of the dielectric sheath layer facilitate the formation of a potential difference between the electrodes and the surrounding tissue, thereby inducing ES at the wound site. The ES intensity is determined by the dielectric properties of the sheath (Fig. 2a). The circuit diagram illustrating the electrical connection between the human body, BET-suture and an ambient EM wave source is displayed in Fig. 2b. Utilizing this circuit diagram, the wound ES potential difference ($V_{incision}$) can be calculated by measuring the body-coupled voltage ($V_b$) and the lagging-induced potential of the

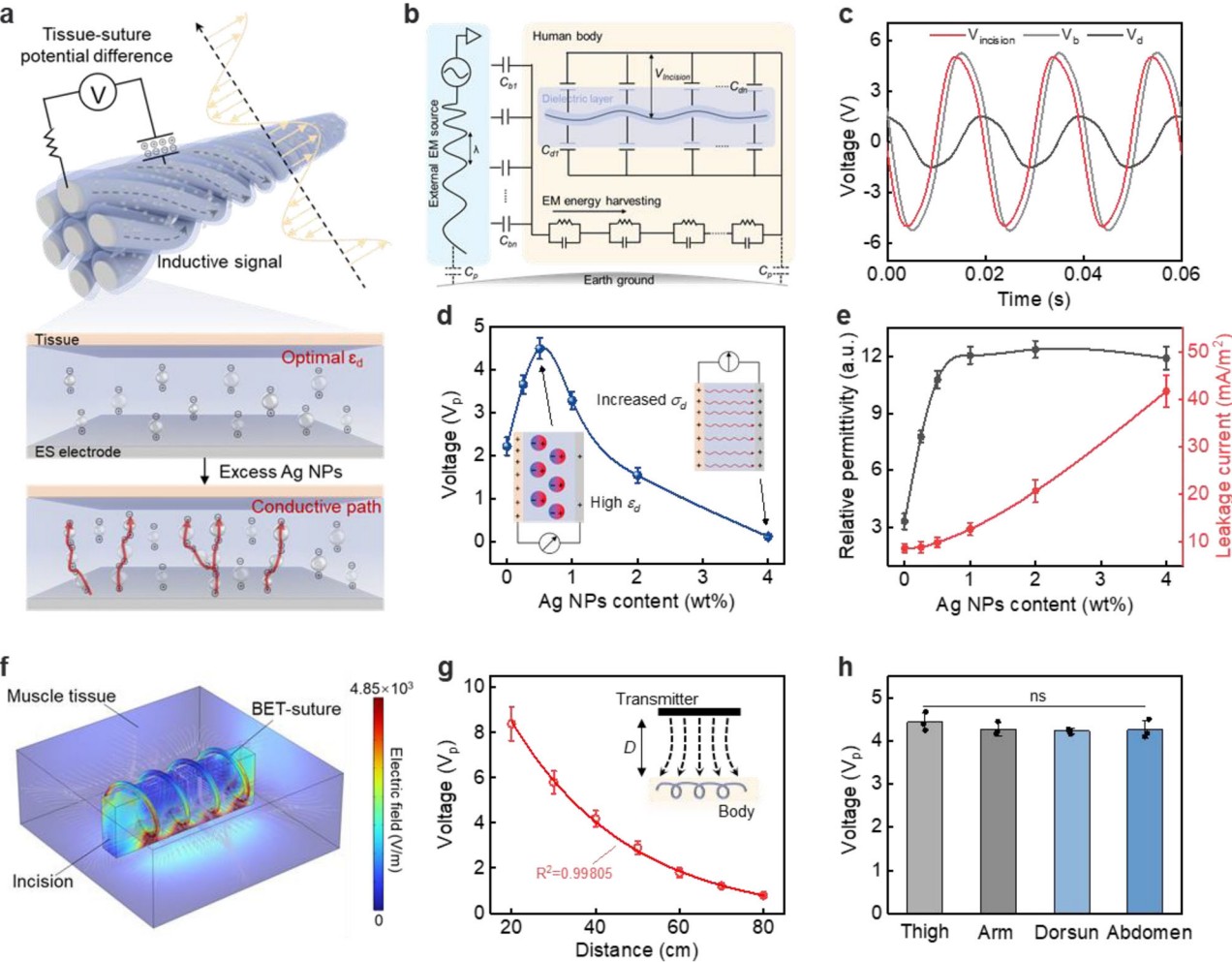

**Fig. 2 | Mechanism of electrical stimulation based on dielectric voltage difference. a** Formation of tissue-suture potential difference and effect of Ag NPs content on the dielectric layer. Appropriate Ag NPs enhance the dielectric properties, while excessive addition of Ag NPs will form certain conductive paths. $\varepsilon_d$, relative permittivity of BET-suture's sheath layer. **b** Equivalent circuit diagram of BET-suture implementing ES by body-coupled EM energy. $C_b$ body-coupled capacitance, $C_d$ capacitance of BET-suture's dielectric layer, $C_p$ parasitic capacitance between body and earth ground, $V_{incision}$ electric potential difference between body and BET-suture's core layer. **c** Real-time measurements of $V_{incision}$, $V_b$, and $V_d$ during wireless ES. $V_b$ open-circuit voltage of body-coupled EM energy, $V_d$ open-circuit voltage of BET-suture's core layer. **d** Peak voltage ($V_p$) of $V_{incision}$ with different Ag NPs contents. $\sigma_d$, conductivity of BET-suture's dielectric layer. $n = 3$ independent samples. **e** Relative permittivity and leakage current density of dielectric layer with different Ag NPs contents. $n = 3$ independent samples. **f** Finite element simulation of E-field strength generated by BET-suture at incision site. The spiral line represents BET-suture and the rectangle represents incision. $V_p$ of $V_{incision}$ (**g**) at different distances from the transmitter and (**h**) at different sites. $n = 3$ independent samples. All statistical analyzes were performed by one-way ANOVA, data represent mean ± standard deviation, ns indicates not significant.

electrodes ($V_d$). The precise calculation procedure is outlined in Supplementary Note S3. The ambient EM field parameters were fixed, and the BET-suture was stitched in the muscle of rats (Supplementary Fig. S8a, Supplementary Note S1). The peak voltages ($V_p$) of $V_b$ and $V_d$ measured in real time were 5.25 V and 1.51 V, respectively, and the calculated $V_p$ of $V_{incision}$ was 5.04 V (Fig. 2c). In contrast, we stitched pure electrodes (no sheath layer) to the same site in another rat to illustrate the criticality of the dielectric layer. The absence of the dielectric layer resulted in the suture failing to establish capacitive contact with the body-coupled EM energy, instead making direct ohmic contact with the electrode (Supplementary Fig. S8b). The $V_b$ and $V_R$ measured in real time were essentially the same, with no significant hysteresis occurring. The calculated $V_{incision}$ was found to be almost 0 V, thereby losing the ES effect (Supplementary Fig. S8c).

In order to enhance the efficiency of ES effects, further improvements were made to the dielectric properties of the sheath layer. The addition of conductive nanofillers to polymers with low dielectric constants has been shown to effectively increase the relative permittivity ($\varepsilon_d$) of the composites[28]. It is further noted that a high dielectric constant of the sheath layer will enhance the $V_{incision}$ (Supplementary Note S3). The enhancement of ES efficiency was achieved by incorporating Ag NPs into the PLGA matrix. With the addition of Ag NPs, the $V_{incision}$ increased and then decreased, with the highest $V_p$ at a content of 0.5 wt% (Fig. 2d, Supplementary Fig. S9). This phenomenon is attributed to the balance between the high $\varepsilon_d$ and the conductivity ($\sigma_d$) of the dielectric sheath. The addition of Ag NPs led to high $\varepsilon_d$, but too much Ag NPs resulted in increased leakage current paths at the interfacial defects between PLGA and Ag NPs, weakening the capacitive properties of the sheath and decreasing the $V_{incision}$ (Fig. 2e, Supplementary Fig. S10). In addition, we have provided relevant parameters and properties of Ag NPs (Supplementary Fig. S11), and discussed in Supplementary Note S5 that the effects of different sizes of Ag NPs and silver nanowires on the dielectric properties of the sheath layer, as well as the advantages of selecting Ag NPs. These results serve to further substantiate the pivotal function of the high-performance dielectric sheath layer in the context of BET-suture.

To provide a more robust demonstration of the ES effect of BET-suture, finite element analysis (FEA) was employed to validate the process. The COMSOL simulation results demonstrated that the dielectric voltage difference formed between the BET-suture and the tissue could create an endogenous E-field at the wound to implement ES (Fig. 2f, Supplementary Fig. S12). Subsequently, the effect of the distance between the BET-suture and the EM field on the ES intensity was explored. As the distance increased, the EM wave loss in spatial transmission increased, resulting in a rapid decline in the $V_{incision}$ and $I_{incision}$ (Fig. 2g, Supplementary Fig. S13). The voltage and current were at the volt and microampere levels, satisfying ES requirements and ensuring personal safety. The average E-field strength generated by the simulation at different distances was calculated and counted (Supplementary Fig. S14a). The results demonstrated that beyond 60 cm, the average E-field strength was below the threshold for effective ES (>0.75 V/mm, as measured in part 2.4). By adjusting the distance appropriately, the ES intensity can be maintained within the effective range. Conversely, the removal of the dielectric layer did not generate an E-field at the incision, which is consistent with the previous result of almost 0 V in $V_{incision}$ calculated for pure electrodes (Supplementary Fig. S14b). Furthermore, $V_{incision}$ decreased with increasing frequency, especially above kHz where the decrease was more pronounced. This is due to the substantial decline in the impedance of the sheath layer at elevated frequencies. Consequently, the body-coupled EM energy can directly traverse the dielectric layer into the electrodes, thereby hindering the generation of a potential difference at the wound site (Supplementary Fig. S15). In addition, BET-suture has been employed to stitch diverse postoperative sites, providing beneficial ES. In rats, the BET-suture was stitched to different sites and the relevant electrical parameters were measured in real time (Fig. 2h, Supplementary Fig. S16). The results showed that $V_b$, $V_d$, and $V_{incision}$ of each site remained almost the consistent, demonstrating the versatile therapeutic properties of BET-suture.

## Charge-enhanced antimicrobial effect based on capacitive properties

In order to achieve wound repair integrity, it is necessary to regulate attenuation of early inflammation that may be triggered by bacterial infection. BET-suture provided capacitive antimicrobial properties that were distinct from the chemicidal anti-inflammatory effect[29]. This is derived from the charge storing capacity of the sheath layer. The embedding of Ag NPs in the sheath layer enabled efficient charge storage during polarization. The presence of a significant number of charges interfered with electron transfer in the bacteria, increasing the production of reactive oxygen species (ROS) and hindering bacterial growth (Fig. 3a). The experimental procedure involved the utilization of a function generator, which was connected to the dielectric sheath layer. A constant input signal (5 $V_{rms}$, 50 Hz) was utilized, and the stored charge of the dielectric layer was measured using an electrostatic meter (Fig. 3b). The PLGA exhibited a rather low charge storage (0.11 $\mu C/cm^2$), while the charge storage of the PLGA/Ag NPs reached a maximum of 5.07 $\mu C/cm^2$ after a 100 s of input (Fig. 3c). Similarly, the KPFM results confirmed the substantial increase in the surface potential of PLGA/Ag NPs (Fig. 3d). To further evaluate the capacitive properties of the dielectric layer, a three-electrode configuration was employed, with a PBS buffer serving as the electrolyte (Supplementary Fig. S17a). The results of the cyclic voltammetry (CV) curves demonstrated that the PLGA/Ag NPs exhibited significant electric double layer (EDL) capacitance behavior, while PLGA exhibited no significant capacitance characteristics (Supplementary Fig. S17b). Furthermore, the capacitance and transient transferred charge density of PLGA/Ag NPs were found to be higher than those of PLGA at different effective voltage values (Supplementary Fig. S17c, d). These results indicated that the sheath layer of BET-suture had significant capacitance characteristics, with the capacity to store a substantial amount of charge

during operation, which could potentially modulate the inflammatory phase.

To validate the capacitive antibacterial properties of BET-suture, bacteria were treated for 3 h using energized BET-suture (BET group), energized BET-suture without Ag NPs (BET no-Ag NPs group), and unenergized BET-suture (BET unenergized group). The bacteria were then collected for incubation and counted (Supplementary Fig. S18a). The BET group exhibited the lowest bacterial culture counts compared to the other groups, indicating that the capacitive antimicrobial properties of BET-suture presented superior bacterial inhibition in comparison to the purely E-field effect (Supplementary Fig. S18b, c). Subsequently, the treated bacteria were subjected to fluorescence staining to monitor bacterial viability and intracellular ROS levels. $H_2O_2$ at a concentration of 0.1 mM was used as a ROS-positive reference, given its ability to kill bacteria by triggering oxidative stress[30]. As demonstrated in Fig. 3e, S. aureus and E. coli exhibited robust growth in the BET unenergized group, revealing that the Ag NPs embedded within the PLGA were unable to reach the bacteria and thus demonstrated suboptimal antimicrobial properties. It is noteworthy that, despite being energized, the BET group demonstrated superior antimicrobial performance compared to the BET no-Ag NPs group, with mortality rates of 76.1% and 79.7%, and 25.1% and 17.9% for S. aureus and E. coli, respectively (Fig. 3f). This indicated that the enhanced antibacterial effect was not due to ES, but rather attributed to the increased charge storage and release capacity of the Ag NP-containing dielectric sheath. This finding further demonstrated that the high dielectric storage charge played a pivotal role in the antibacterial process. Furthermore, the presence of ROS-positive signals in S. aureus and E. coli on the BET group was found to be analogous to those observed in the 0.1 mM $H_2O_2$ group, indicating the induction of oxidative stress during the antimicrobial process subsequent to energization (Fig. 3g). Collectively, these observations provide substantial evidence that BET-suture can effectively eliminate bacteria through the induction of oxidative stress via capacitive antibacterial action.

Antimicrobial action favors the elimination of bacteria from the early healing stage to reduce the intensity of the inflammatory response. However, as the inflammatory response progresses, the electrotherapeutic action of BET-suture is expected to drive the dynamic evolution of the inflammatory phase and the transition to the proliferative phase. M1 macrophages primarily orchestrate host defences and promote the secretion of pro-inflammatory cytokines, which exacerbate tissue damage and disease progression. In contrast, M2 macrophages promote the secretion of anti-inflammatory cytokines for active wound repair[13]. ES has been shown to possess significant inflammatory scavenging capacity, inducing a shift from M1 to the M2-type macrophages (Supplementary Fig. S19a). The enzyme-linked immunosorbent assay (ELISA) experiments demonstrated that three pro-inflammatory factors (TNF-α, IL-1β, and IL-6) derived from M1 macrophages in the BET group exhibited significantly lower expression levels compared with the BET no-Ag NPs group, while the anti-inflammatory factor (IL-10) from M2 macrophages showed higher expression levels (Supplementary Fig. S19b). These results demonstrated that the capacitive (stored charge) electrotherapy provided by BET-suture not only achieved superior antimicrobial properties, but also resulted in more efficient clearance of inflammatory cytokines, aiding wound passage through the inflammatory phase quickly.

## Threshold of electrical stimulation intensity and cell signaling pathways

ES has been proved to modulate the proliferation and remodeling phases of wound repair and accelerate tissue healing[15]. However, current ES treatment modalities either necessitate a power supply or are performed using ultrasound or force-driven power generation devices (Fig. 4a, Supplementary Table S2). The first two types of devices consume more than $10^2$ $\mu W$ of power during therapy, which is significantly

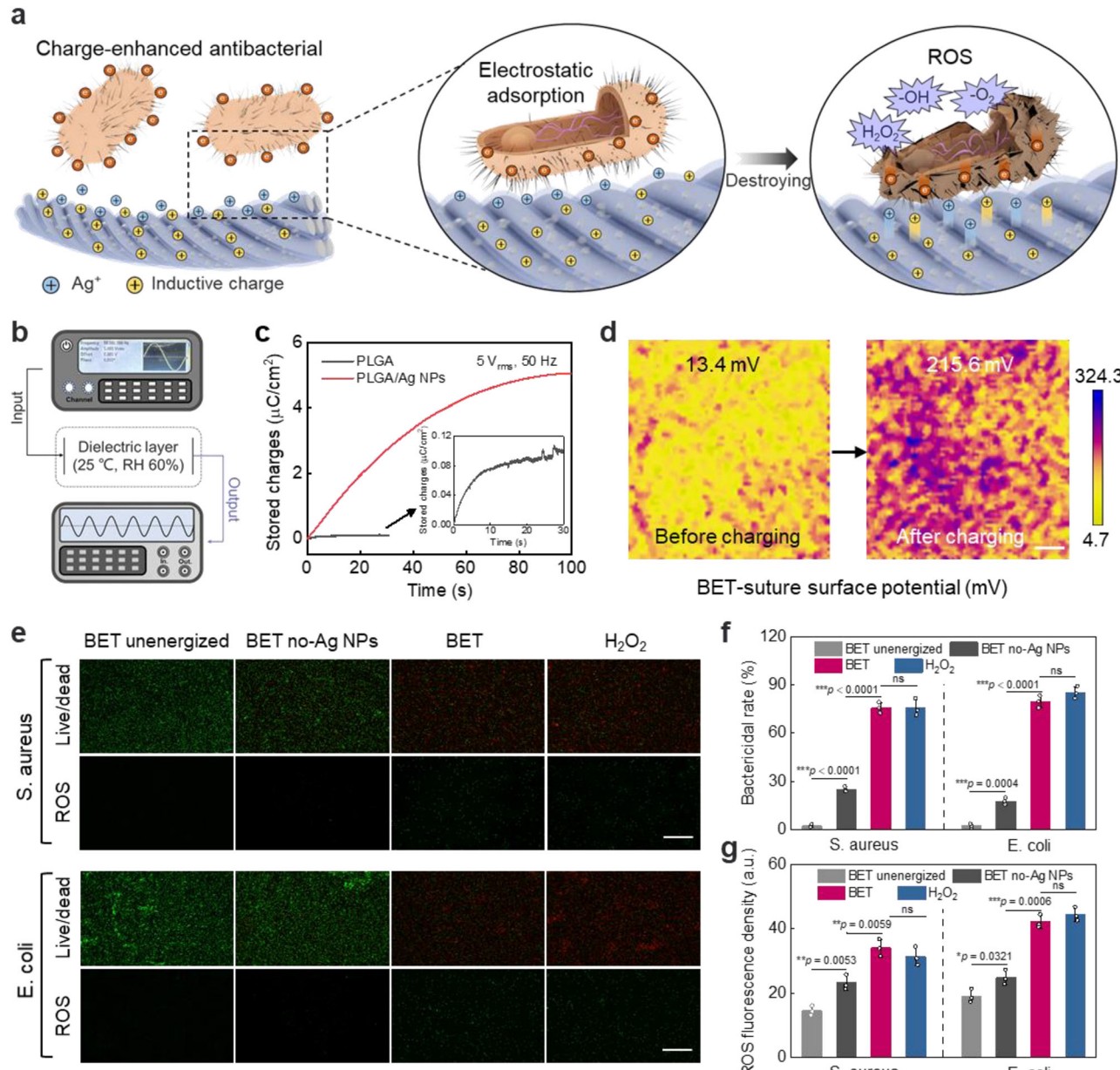

**Fig. 3 | Charge-enhanced antimicrobial effect based on capacitive properties.** **a** Diagram of the antibacterial mechanism of BET-suture. The polarized dielectric layer inspired instantaneous electron transfer between the bacteria and the surface of BET-suture. The enormous electron loss and the action of silver nanoparticles induced ROS burst within bacteria to realize the antibacterial properties. **b** Diagram of experimental setup for measuring the charge storage capacity of the dielectric layer and **c** the relationship between the stored charges and time. Effective voltage 5 $V_{rms}$, frequency 50 Hz. **d** KPFM images depicting the surface potential of dielectric layer before and after polarization. Scar bar: 30 μm. n = 3 independent samples. **e** Live/dead and ROS staining images of S. aureus and E. coli treated with unenergized BET-suture, energized BET-suture without Ag NPs, energized BET-suture and 0.1 mM $H_2O_2$. Scar bar: 50 μm. **f** Quantitative analysis of the live/dead staining results and **g** ROS fluorescence intensity of treated bacteria. $n = 3$ independent samples. All statistical analyzes were performed by one-way ANOVA, data represent mean ± standard deviation, ***$p < 0.001$, **$p < 0.01$, *$p < 0.05$, ns indicates not significant.

---

higher than force-driven ES devices and our BET-suture. But force-driven ES devices, due to their mechanoelectric conversion mechanism, offer an almost arbitrary and uncontrollable stimulation intensity, with the potential to result in suboptimal therapeutic outcomes. BET-suture overcomes these limitations by enabling wireless stimulation with effective intensity through the modulation of environmental parameters (EM field or distance). A range of ES intensity was explored through fibroblast migration and proliferation experiments (Supplementary Fig. S20a). Notably, BET-suture demonstrated no discernible heating even after two hours of uninterrupted operation, ensuring the stability of the stimulation process following the stitching procedure

(Supplementary Fig. S20b). The ES intensity to which the cells were subjected in the experiments was determined by FEA (Fig. 4b, Supplementary Fig. S21). Fig. 4c showed that ES effectively accelerated the migration of fibroblasts at average E-field strengths in the range of less than 5 V/mm (Supplementary Fig. S22). Conversely, the migration of fibroblasts was found to be inhibited by the application of E-field strengths that exceeded 5 V/mm. Consistent with these observations, the cell proliferation assay revealed a substantial decrease in fibroblast proliferation at elevated E-field strengths (Fig. 4c). To understand the reason for the inhibition of cell behavior at high E-field strength, cell live/dead staining was performed for each group. The results showed

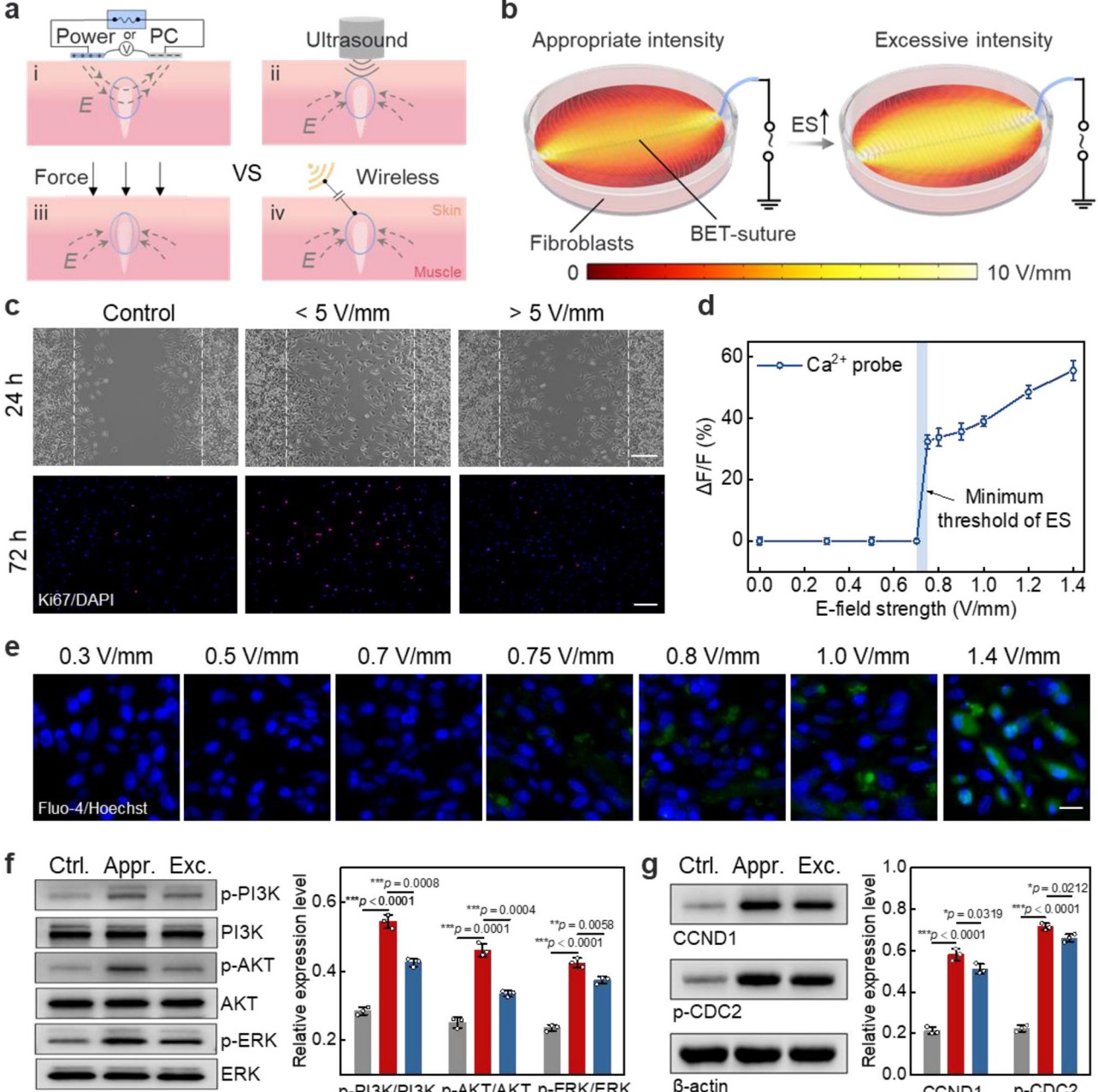

**Fig. 4 | Threshold of electrical stimulation intensity and cell signaling pathways. a** The current mainstream ES techniques, including i) direct energy-fed ES, ii) ultrasound-driven ES, and iii) mechanoelectric conversion ES, are compared with the principles of the iv) wireless ES developed in this work. **b** Schematic of the in vitro experimental setup of BET-suture applying different intensity of E-field. The middle insets show the finite element simulation results of E-field distribution via BET-suture electrodes. **c** Microscopic images after 24 h of the scratch healing assay and Ki67 fluorescence staining images after 72 h of the cell proliferation assay. Scar bar: 80 μm. $n = 3$ independent samples. **d** Plot of relative fluorescence intensity of intracellular $Ca^{2+}$ probe against the applied ES voltage. $n = 3$ independent samples. **e** Representative fluorescence images of activated $Ca^{2+}$ fluorescent probe in fibroblasts after ES action with applied voltages ranging from 0 to 140 mV. Scale bar: 30 μm. $n = 3$ independent samples. Western blot to detect **f** the phosphorylation levels of PI3K, AKT and ERK and **g** relative expressions of CCND1 and p-CDC2 compared to β-actin in fibroblasts under ES. $n = 3$ independent samples. All statistical analyzes were performed by one-way ANOVA, data represent mean ± standard deviation, ***$p < 0.001$, **$p < 0.01$, *$p < 0.05$.

that ES at high E-field strength killed part of the cells, leading to significant cellular damage (Supplementary Fig. S23). Consequently, to ensure the safety and efficacy of the treatment, the maximum threshold of ES intensity applied by BET-suture should be less than 5 V/mm.

Changes in cellular behavior under ES are usually triggered by calcium influx, driven by the opening of voltage-gated calcium channels (VGCC)[31]. The present study explored this mechanism by using

$Ca^{2+}$ fluorescent probes and obtained the minimum threshold for inducing cell proliferation and migration behavior. Fluorescence staining results showed that stimulation using an E-field strength of 0.75 V/mm started to induce $Ca^{2+}$ endocytosis (green), whereas there was no significant signal at 0.7 V/mm (Fig. 4e). Statistical analysis of relative fluorescence intensity revealed a pronounced increase in $Ca^{2+}$ signal intensity between 0.7 and 0.75 V/mm, reaching a significant level at 1.4 V/mm (Fig. 4d). These results suggested that the minimum

threshold of effective ES intensity required for the BET-suture to function was 0.75 V/mm. As previously mentioned, the ES intensity of the BET-suture could be flexibly adjusted by environmental parameters to cover this effective interval, thereby ensuring optimal ES performance.

Calcium influx tends to trigger the expression of downstream relevant signaling pathways to activate the cells. Subsequent examination of the alterations in cellular metabolism in response to ES was undertaken to elucidate the corresponding action mechanisms. The ELISA results demonstrated a significant increase in the secretion of three related growth factors, including epidermal growth factor (EGF), transforming growth factor-β (TGF-β), and vascular endothelial growth factor-A (VEGF-A), under effective ES (Supplementary Fig. S24). Then, we detected the related protein signals downstream of growth factor expression by western blot (WB). The PI3K and MAPK signaling pathways are closely linked to the expression of growth factors and to function as pivotal signaling pathways during cell proliferation. The WB results showed a significant increase in the phosphorylation of proteins involved in the action of the signaling pathways (PI3K, AKT, and ERK), which indicated the activation of the signaling pathways (Fig. 4f). This demonstrated that not only was growth factors secretion increased but also higher expression was induced under effective ES.

The activation of cell signaling pathways must induce changes in the cell cycle to trigger the onset of proliferation. The relative expression of cell cycle protein D1 (CCND1) and phosphorylated-CDC2 (p-CDC2) were measured by WB. CCND1 is a G1/S-phase-related protein involved in the process of DNA replication; p-CDC2 is a G2/M-phase-related protein involved in the process of mitosis. The results showed that ES significantly augmented the expression of CCND1 and p-CDC2 in fibroblasts, signifying optimal cell cycle function (Fig. 4g). Similarly, the ELISA and WB results obtained under high intensity ES were consistent with the results of cell migration and Ki67 staining experiments, i.e., cell damage resulted in suppression of the relevant expression (Fig. 4f, g, Supplementary Fig. S24). The alterations in the overall behavior of the cells under ES were demonstrated in full detail in Supplementary Fig. S25. Under effective ES (75–500 mV/mm), calcium channels were activated and enhanced the secretion and expression of cell-associated growth factors, which in turn activated cell cycle behaviors through the PI3K and MAPK signaling pathways and promoted wound healing.

## Evaluation of incision healing effect in vivo

To demonstrate the comprehensive modulation of the wound repair phase by the electrotherapeutic actions (effective ES and capacitive antimicrobial & anti-inflammatory synergism) of BET-suture, we established a rat muscle wound model to validate the in vivo therapeutic effect (Fig. 5a). The specific schematic timeline of the experimental protocol is shown in Supplementary Fig. S26a. After 7 days of stitching, four groups of repaired tissues were removed for observation: the medical absorbable suture (control) group, the antimicrobial suture (AB-suture) group, the electrical stimulation suture (BS-suture) group, and the BET-suture group. The muscle tissue with BET-suture stitched did not show obvious incision and had the most optimal recovery status (Fig. 5b). Furthermore, the tissue sections of the removed muscle samples were subjected to staining (Fig. 5c). The hematoxylin eosin (H&E) staining revealed that the muscle tissue had almost completely recovered in the BET-suture group, while the other three groups exhibited unrepaired incisions with varying degrees of inflammation. A detailed comparative analysis of the recovery status and anti-inflammatory properties of the three different suture materials is provided in Supplementary Note S6. Masson staining revealed significant collagen deposition and alignment in the BET-suture group, indicating almost complete remodeling of the muscle tissue.

This desirable repair behavior can be attributed to the stable and effective electrotherapeutic properties of BET-suture during the repair period (Fig. 5d, Supplementary Movie S3), its enhanced EMG signal generation (Supplementary Fig. S26b, c), and its superior repair efficiency, which was 1.43 times higher than that of ES-suture (Fig. 5e). Subsequent continuous monitoring of the effective action of BET-suture in vivo and analysis of the trend of electrical performance is documented in Supplementary Note S7. After 28 days, BET-suture lost its ES effect due to complete degradation of the sheath (Supplementary Fig. S27a, b); however, the presence of the electrodes was still able to provide high strength to close the wound (Supplementary Fig. S27c). Subsequently, we collected tissues from vital organs (including lung, liver, spleen, kidney, and heart) and performed histological examination by H&E staining (Supplementary Fig. S28). The results showed that, compared to normal rat organs, the rat organs stitched with BET-suture did not exhibit signs of pathological inflammation or systemic immune responses, such as abnormal lymphocyte infiltration in vital organs, thereby ruling out the risk of functional impairment and organic lesions. Furthermore, even when the wound was in an inflammatory or fibrotic state, it did not affect the ES performance of the BET-suture (Supplementary Fig. S27d).

Subsequently, the status of tissue repair was assessed using immunofluorescence staining. Staining for α-smooth actin (α-SMA), a representative molecule of myofibroblast transformation, demonstrated that the ES-suture group exhibited the highest rate of positive staining, while the BET-suture group exhibited a slightly lower rate of staining (Supplementary Fig. S29). Conversely, the staining results for the vascular endothelial cell marker CD31 demonstrated that the BET-suture group exhibited higher expression (Fig. 5f, g). These results suggested that myofibroblasts in the BET-suture group have been partially deactivated and transformed into quiescent fibroblasts, and that vascular proliferation would gradually plateau, indicating that the repair stage was in the remodeling phase. The presence of inflammatory infiltration was further observed using CD3- and CD20-labeled T cells and B cells, respectively (Supplementary Fig. S30). The results showed that, with the exception of the BET-suture group, the other three groups exhibited different degrees of inflammatory infiltration. When these results were combined with the α-SMA and CD31 immunofluorescence results, it was indicated that the other three groups were still at a certain stage of the inflammatory-proliferative phase. This further demonstrated the ability of BET-suture to comprehensively regulate wound healing.

Furthermore, immunohistochemistry (IHC) was utilized to observe the cellular metabolic changes in the proliferation-remodeling phase. The results showed that the growth factors (EGF, TGF-β, VEGF-A) of the BET-suture group exhibited elevated secretion (Fig. 5h, Supplementary Fig. S31). Concurrently, proteins associated with PI3K and MAPK signaling pathways demonstrated heightened phosphorylation (Supplementary Fig. S32), validating the expression of growth factors. In addition, to ascertain that BET-suture facilitated the transition of the wound from the early inflammatory phase to the proliferative phase, we removed tissue samples on day 3 of healing and assayed the expression of relevant cytokines. The results demonstrated that several pro-inflammatory cytokines (TNF-α, IL-6, and IL-1β) released by M1 macrophages were downregulated by BET-suture, whereas the anti-inflammatory cytokine (IL-10) released by M2 macrophages was significantly upregulated (Fig. 5i). This behavior of high expression of growth factors and low secretion of pro-inflammatory factors suggested that the wound may have entered the proliferative phase by day 3. Overall, these results corroborate the aforementioned conclusion that BET-suture can inhibit the inflammatory response and rapidly regulate the wound to pass through the inflammatory phase via wireless electrotherapeutic modalities. The subsequent enhancement of secretion and expression of relevant growth factors during the proliferation phase has been demonstrated to promote angiogenesis and extracellular matrix deposition, thereby accelerating tissue remodeling.

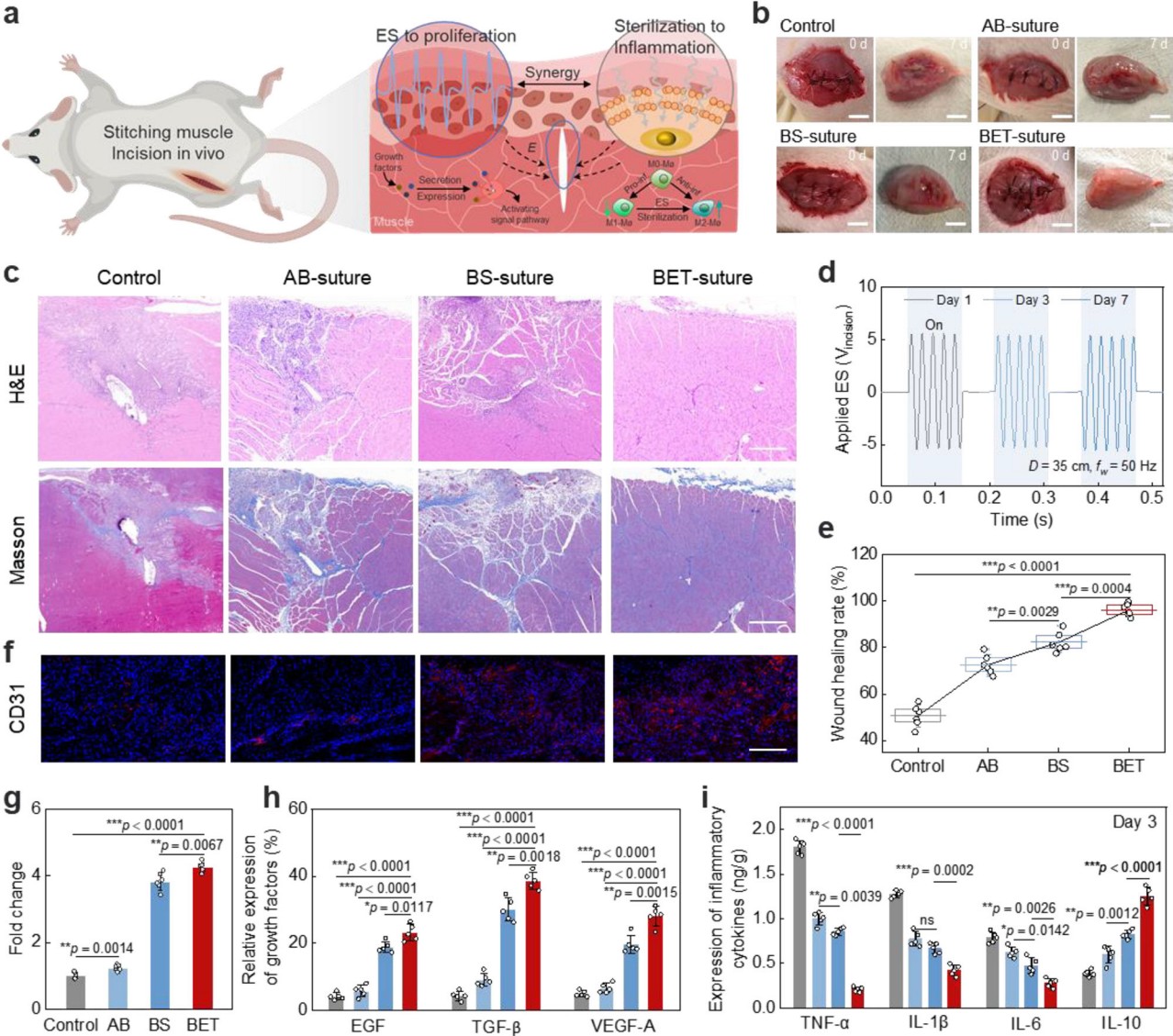

**Fig. 5 | Evaluation of incision healing effect in vivo. a** Schematic diagram of BET-suture-based ES and capacitive antimicrobial synergistically assisting incision repair in vivo. **b** Optical photographs of the incisions stitched in each group and the incisions and their surrounding muscle tissues removed in each group after 7 days of stitching. Scale bar: 2 cm. **c** H&E and Masson staining images of the incision and its surrounding tissues in each group. Scale bar: 500 μm. $n = 6$ independent samples. **d** Real-time signals of $V_{incision}$ continuously monitored over 7 days of stitching. **e** Wound healing rate of each group calculated by section staining. The box represents the 25–75% confidence interval, the triangle represents the mean, the error bars represent the standard deviation, and the horizontal line in the box represents the mean. $n = 6$ independent samples. **f** CD31 immunofluorescence staining of healing tissues in each group and **g** its quantitative analysis. Scale bar: 200 μm. $n = 6$ independent samples. **h** Quantitative analysis of growth factors (EGF, TGF-β, VEGF-A) immunohistochemical staining results of the removed tissues. $n = 5$ independent samples. **i** Cytokines (TNF-α, IL-1β, IL-6, IL-10) expression levels of the wound tissues after 3 days of stitching. $n = 5$ independent samples. All statistical analyzes were performed by one-way ANOVA, data represent mean ± standard deviation, ***$p < 0.001$, **$p < 0.01$, *$p < 0.05$, ns indicates not significant.

## Discussion

In conclusion, we firstly reported a body-coupling-electrotherapy suture (BET-suture) with wound immobilization, full-stage healing regulation, sustainability and low energy consumption. The BET-suture has been shown to regulate the tissue repair phases in a synergistic manner: it generates an effective stimulating E-field at the wound by converting body-coupled EM energy and induces capacitive antimicrobial properties to further enhance the healing efficiency. The BET-suture demonstrated comparable strength to ligament sutures (Supplementary Table S1), while exhibiting the flexibility of commercial surgical sutures. Furthermore, the BET-suture could be resorbed or expelled from the body. Adaptive electrical properties enabled BET-suture to remain in the effective stimulation zone during repair. In vivo wound models have demonstrated that BET-suture promoted a rapid

transition through the inflammatory phase, accelerated tissue remodeling and reduced postoperative complications, including hyperplasia and infection. In comparison with existing monofunctional sutures, BET-suture comprehensively addressed the role of relevant cytokines and signaling pathways at different restorative stages (Supplementary Table S3), providing the desirable therapeutic capability for postoperative wounds. In the future, we look forward to applying BET-suture to promote wound healing in other tissues or organs and to achieving clinical translation.

## Methods
### Materials
Mo filament (50 μm) was bought from Xi'an Huanyu Tungsten & Molybdenum New Material Technology Co, Ltd., China. Poly(lactic-co-

glycolic acid) (PLGA, 75:25) masterbatch was purchased from Nature Works, USA. Silver nanoparticles (Ag NPs, 10–500 nm) was purchased from Shanghai Meiyi Alloy Materials Co. Ltd., China. Hexafluoroisopropanol (HFIP) was purchased from Shanghai Aladdin Reagent Co. Ltd., China. The synthetic skin (food-grade silicone) was purchased from Guangzhou Huijin Science and Education Model Co., Ltd.

## Fabrication of BET-suture

Firstly, the PLGA/Ag NPs composite spinning solution was prepared. The PLGA masterbatch was dissolved in HFIP with a concentration of 8 wt% and stirred at room temperature for 4 h to obtain a transparent PLGA spinning solution. Then Ag NPs with different mass ratios (0.25 wt%, 0.5 wt%, 1.0 wt%, 2.0 wt% and 4.0 wt%, representing the mass percent of Ag NPs in the PLGA matrix) were added and stirred for 2 h until well-mixed to obtain PLGA/Ag NPs composite spinning solution. Secondly, the dry pulling spinning technique was used to prepare primary fiber (Mo@PLGA/Ag NPs). Mo filament was drafted into the PLGA/Ag NPs composite spinning solution and directed through the spinning head (inner diameter of 80 μm) at a stretching rate of 2 m/min to the heating chamber for rapid removal of the HFIP solvent, resulting in the core-sheath primary fiber of PLGA/Ag NPs-coated Mo filament. Thirdly, the multi-stranded (5, 8 and 11 strands) primary fibers were twisted using a doubling and twisting combined testing machine (Anytester AT208, China) with consistent degree of twist to obtain BET-suture.

## Materials characterization

Field emission scanning electron microscopy and EDS (FESEM, S-4800, Japan) was used to characterize the microscopic morphology and elemental distribution. The Ag NPs were dispersed in a deionized water/ethanol solution, and the Zeta potential and particle size distribution of the Ag NPs were measured using dynamic light scattering (DLS, Malvern Zetasizer Nano ZS90, UK). Mechanical properties of BET-suture were tested by an electronic universal testing machine (Instron 5969, from Instron Corporation, USA) at a tensile rate of 100 mm/min. A sheet of $2 \times 2$ cm PLGA/Ag NPs composite film of the same thickness (15 μm) as the sheath layer was prepared for measuring physical properties. Specifically, LCR meter (E4980A, Agilent Technology, USA) and broadband dielectric impedance spectrometer (Concept 40, Labmates Technology, China) were used to measure capacitance at 50 Hz, relative permittivity and impedance in the frequency range of $1–10^7$ Hz, respectively. An electrochemical workstation (Biologic SAS, VSP-300, France) was used to measure CV curves and leakage current density. Kelvin probe force microscopy (Park NX20, South Korea) was applied to measure the surface potentials. The in vitro degradation behavior of BET-suture was studied using a thermostatic shaking incubator (HZQ-F100, China) with the following conditions: PBS buffer (pH = 7.4), 37 °C. Fresh PBS was changed twice a week. The tissue drag force per unit circumference was measured using a tensile tester (Sanliang, SMF-50, Japan) when the suture was pulled through synthetic skin, rat skin, and rat muscle.

## Electrical characterization

The Keithley 6514 (from Keithley Instruments, USA) and oscilloscope (Keysight DSOX3012T, USA) in conjunction with a 1 megaohm probe were used to test the voltage signals of BET-suture in vitro and in vivo. The wireless signal sources of specific conditions were realized by connecting the RF/microwave signal generator (MODEL 835) to the power amplifiers (LZY-22+)/(ATA-7050, Agitek). COMSOL Multiphysics (v.6.1) was used to simulate E-field strength at the cell culture dish or incision site performed by the BET-suture. Unless otherwise specified, the optical/electrical signals of the manuscript were measured at a distance of 30–40 cm from the transmitter (23.8 dBm, 50 Hz).

## Biocompatibility test

All materials of BET-suture were tested for biocompatibility. L929 fibroblasts (Chinese Academy of Science Cell Bank, Shanghai, China) were cultured in DMEM-HG medium containing 10% FBS and 1% penicillin/streptomycin. The cell culture conditions were humid incubator with 5% $CO_2$ at 37 °C. The L929 fibroblasts were seeded into 96-well plates, 200 μL per well, and placed in a constant temperature incubator for 24 h. After removing the culture medium, three samples (Mo filament, Mo@PLGA fiber and BET-suture) were cut into 1 cm segments, and cells in different wells were separately added different sutures and different Ag NPs concentrations (10 ppb and 100 ppb). After 72 h, 22 μL of Cell Counting Kit-8 (CCK-8, Solarbio, China) was added to the wells to be tested, and then incubated at 37 °C for 1 h. The absorbance was measured at 450 nm to assess cell activity according to the optical density (OD) values.

The biocompatibility of BET-suture and Ag NPs of different sizes (50, 200, 500 nm) and Ag NWs (diameter 90 nm, length 2–20 μm) on L929 fibroblasts was also tested with Live/Dead staining. Briefly, Calcein/PI staining was employed to distinguish live cells from dead cells, and then these samples were observed under a fluorescence microscope (BX-51, Japan) after incubation for 15 min.

## Antibacterial activity

Staphylococcus aureus (S. aureus, ATCC 43300) and Escherichia coli (E. coli, ATCC 25922) were cultured in tryptic soy broth (TSB) medium for 24 h at 37 °C. Each bacterial culture was divided into three groups, where unenergized BET-suture, energized BET-suture without Ag NPs, and energized BET-suture in the respective cultures ($1 \times 10^8$ CFU/mL). The electrical parameters for the BET group and the BET no-Ag NPs group were 50 Hz and 5 $V_{rms}$. After incubation at 37 °C for 3 h, diluted 100 μl of bacterial solution was spread evenly on agar culture plates. The plates were incubated at 37 °C for 24 h and bacterial colonies were counted. In addition, the treated bacteria after incubation at 37 °C for 3 h were then stained in the dark for 30 min with a Live/Dead™ BacLight™ Bacterial Viability Kit (Invitrogen, USA) containing propidium iodide (PI, 3 μL/mL) and Syto-9 (1:1). The stained bacteria were placed on microscope slides, and the fluorescence of live and dead bacteria was observed using a fluorescence microscope (Ts2R, Nikon, Japan). The fluorescence intensity of dead bacteria was measured using Image J software (NIH, Bethesda, MD, USA).

The intracellular ROS levels were determined by the fluorescent probe, 2′, 7′-dichlorodihydrofluorescein diacetate (DCFH-DA, Beyotime, China) which could be deacetylated and oxidized to fluorescent products after crossing the membrane of live bacteria. 400 μL of DCFH-DA were spread on the treated bacteria surface with protection from light for 15 min. The excess dye was removed by PBS and the samples were put on the sample stage under an inverted fluorescent microscope with 488 nm as the excitation wavelength and 520 nm as the emission wavelength. The bacteria treated with 0.1 mM $H_2O_2$ served as the ROS positive group.

## Immune regulation

RAW 264.7 cells (Chinese Academy of Science Cell Bank, Shanghai, China) were cultured in a 6-well plate and incubated for one day at a density of $1 \times 10^5$ cells per well. Lipopolysaccharide (Solarbio, 100 ng/mL) was used to induce M1 polarization for one day. After that, the unenergized BET-suture, energized BET-suture without Ag NPs, and energized BET-suture was immersed in DMEM or applying ES for one day (1 h every 12 h), respectively. Cytokine levels in the culture media were quantified using TNF-α, IL-1β, IL-6 and IL-10 rat ELISA kits (Abcam, UK), respectively.

## Threshold study of maximum ES intensity

The migration and proliferation behavior of cells under ES was used to determine the appropriate E-field strength. The migratory behavior of cells stimulated by an E-field was determined using the scratch assay.

L929 fibroblasts were cultured in 6-well cell plates for 48 h. BET-suture was immobilized in a polyimide membrane and placed at the bottom of the petri dish. A pipette tip was used to make scratches of ~400 μm on the fibroblasts covering the 6-well cell plate. The RF/microwave signal generator (MODEL 835) was connected to the BET-suture electrodes and inputted the induced potentials measured in the rats, and FEA was used to determine the distribution of the E-field strengths in the cell culture dish. Cells were intervened at different stimulus intensities and migration of each group was observed after 24 h. ES was performed every 12 h for 1 h.

The proliferative behavior of the cells under ES was further observed. L929 fibroblasts were cultured in 6-well cell plates for 24 h. Fibroblasts in the control group did not receive any intervention. The fibroblasts in the experimental group were treated with ES using different E-field strengths, respectively. ES was performed every 12 h for 1 h. After 72 h of incubation in the cell culture incubator, cells were washed three times with PBS and fixed with 4% paraformaldehyde (Sangon Biotech, China) at room temperature for 20 min. The cells were then permeabilized with 0.05% Triton X-100 (Sangon Biotech, China) for 10 min and blocked with 10% BSA (Solarbio, China) for 30 min. Subsequently, the cells were incubated overnight at 4 °C with rabbit anti-Ki-67 antibody (1:400, Abcam). On the following day, unbound antibodies were washed off by rinsing the cells three times for 10 min each with PBS. The cells were then incubated with Goat Anti-Rabbit IgG H&L (Alexa Fluor® 488) (1:500, Abcam) at room temperature for 1 h. Ki67 was used for immunofluorescence staining and quantified the proportion of Ki67-positive cells and the relative fluorescence intensity of Ki67 in each group. Finally, DAPI (Sigma-Aldrich, USA) was applied to label nuclei and preserve fluorescence. All images were acquired using a confocal microscope (LSM 710, ZEISS, Germany). The percentage of Ki-67 positive cells was determined by randomly counting five fields of view. In addition, cell survival was observed in each group after stimulation using cell live/dead staining. Note that all values of ES strength were taken as the average E-field strength at a distance of 200 μm from the BET-suture.

## Ca²⁺ signal analysis of the lowest ES threshold

The changes in intracellular $Ca^{2+}$ concentration were evaluated using the calcium probe. Fluo-4 AM (Thermo Fisher, USA) was added to the culture medium of L929 fibroblasts to reach a final concentration of 1 μM. The electrical stimulation device (BET-suture encapsulated in a PI film) was placed at the bottom of the cell culture dish and the ES was applied as in the cell migration experimental procedure. A series of 50 Hz E-fields of 0–1.4 V/mm were applied at 0.2 V/mm intervals. In subsequent fine-tuned E-field studies, the 50 Hz E-field of 0.7–0.8 mV was applied at 0.05 V/mm intervals. After incubation at 37 °C for 10 min, cells were washed three times with PBS and fluorescence intensity was quantified. In the calcium signal data presented in this study, relative fluorescence changes ($\Delta F/F_0$) were used. $\Delta F/F_0 = (F − F_0)/F_0$, where F is the current fluorescence intensity of the cell's region of interest, and the baseline fluorescence value $F_0$ is the average of the lowest 25 % of all fluorescence signals.

## Quantification of growth factors

ELISA assay was used to quantify the secretion of growth factors during cell proliferation. Cell culture supernatants from each group of proliferation experiments were collected at different time points (12 h, 24 h and 48 h), and the concentrations of growth factors in the supernatants were detected by rat EGF, TGF-β, and VEGF-A kits, respectively. All ELISA tests were performed according to protocols recommended by vendors. The absorbance at 450 nm was measured by microplate readers.

## Signaling pathway analysis

Western blot (WB) was used to analyze the behavior of signaling pathway activation in response to ES of cell proliferation. Samples were randomly taken after 72 h of cell proliferation and analyzed by protein blotting using 10% sodium dodecyl sulfate-polyacrylamide gel electrophoresis (SDSPAGE) for PI3K, p-PI3K, AKT, p-AKT, ERK, p-ERK, CCND1, p-CDC2. Relevant antibody information is listed in Supplementary Table S4. The membranes were then washed thoroughly with PBS three times, and the chemiluminescence of the bands was developed with a Fast Western Blot Kit (Thermo Fisher, USA) then imaged and quantified using an Imaging Densitometer (Bio-Rad, CA, USA). Gray values were determined using Image J software. Uncropped and unprocessed scans of the blots are visible in the source data file.

## Animal experiments

All animal experiments were performed according to protocols approved by Committee on Ethics of Donghua University (DHUEC-NSFC-2022-43). Purchased from SiPeiFu Biotechnology Co., LTD (Suzhou, China), several six-eight-weeks-old Sprague-Dawley female rats with an average weight of 250–300 g were kept in a certified animal facility. Mice were group-housed if same sex little-mates were available. All the samples were sterilized by UV before initiating the experiments. All rats were housed uniformly in an accredited animal facility at Shanghai Sixth People's Hospital with a 12-h light/dark cycle, controlled ambient temperature of 22 ± 2 °C, and relative humidity of 50–60%, with free access to food and water.

Rats were divided into four groups: (1) control group, where incisions were closed using commercially absorbable (PGLA) sutures; (2) AB-suture group, where incisions were closed using BET-suture but no ES was applied; (3) BS-suture group, where incisions were closed using BET-suture without added Ag NPs and ES was applied; (4) BET-suture group, where incisions were closed using BET-suture to suture the incision and produce a synergistic therapeutic effect. After anaesthetizing the rats with sodium pentobarbital (50 mg/kg), their leg hair was removed with an electric hair clipper. The skin was slashed with a scalpel and an incision of 3 cm long and 1 cm deep was made vertically in the muscle. Then each group was individually stitched to close the muscle incision, and the skin was stitched with medical nylon thread. 1 h of ES was applied to the ES-suture group and the BET-suture group every 12 h. The ES performance of the BET-suture was continuously monitored during the 1–35 days, and the variations in electromyographic (EMG) signals of the rats were recorded using the Neusen.U16 (Changzhou Boruikang Technology Co. Ltd., China). All rats in the study survived and remained normally healthy. After implantation, the corresponding rats were executed on days 3 and 7, respectively. Tissue sections located in the center of the original wound and some of the intact muscle around it were cut from each rat for subsequent pathological analysis.

## Histological, immunofluorescence and immunohistochemical staining, and ELISA assay

Tissue for pathological analysis was excised and fixed in 10% formalin for 24 h. The samples then underwent a general procedure (dehydration, clearing, and embedding) for histopathological analysis. Paraffin sections with 4 μm thickness were processed with H&E and Masson stain. Histology images were collected on an inverted optical microscope. Image J was used to measure areas of the wounds and to analyze the positive expressions. The wound healing rate (η) and the ratio of healing efficiency (Φ) were calculated using the following equations:

$$\eta = \frac{S_0 - S_t}{S_0} \times 100\% \tag{1}$$

$$\emptyset = \frac{\eta_x - \eta_c}{\eta_x} \tag{2}$$

where, $S_0$ represents the initial wound area, $S_t$ represents the wound area after a certain time of healing, $\eta_x$ represents the wound healing

rate of a certain experimental group, and $\eta_c$ represents the wound healing rate of the control group.

Some vital organs including lung, liver, spleen, kidney, and heart were collected from another part of the rats for H&E staining after euthanasia at 28 days post implantation. Tissues were fixed with 10% formalin and slices were prepared at 4 μm for H&E staining.

For immunofluorescence staining, serial tissue sections were processed according to the standard immunofluorescence staining procedures, including deparaffinization, antigen-retrieval (citrated buffer, heat-induced), permeabilization (0.3% Triton PBST), and antigen blocking (goat serum, Cell Signaling Technology, USA). Then, as-prepared sections were incubated with antibodies, including mouse monoclonal α-SMA primary antibody (1:200; Novus Biologicals, USA), rabbit polyclonal CD31 primary antibody (1:200; Abcam, UK), rabbit monoclonal CD3 primary antibody (1:200; Abcam, UK), and mouse monoclonal CD20 primary antibody (1:200; Santa Cruz Biotechnology, USA). Then the sections were PBST-washed twice and incubated with secondary antibodies (goat anti-mouse Alexa 488, donkey anti-rabbit Alexa 555, Thermo Fisher) at room temperature before counterstaining with DAPI. Fluorescent images were captured via Zeiss Axio Observer 5 Inverted Phase Contrast Fluorescent Microscope.

For immunohistochemical staining, 5-μm-thick paraffin-embedded tissue sections removed on days 3 were dewaxed in xylene and rehydrated in graded ethanol series. Antigenic repair was performed by microwave heating. Non-specific antigen was blocked with 1.5% normal goat serum. Slides were incubated overnight with EGF, TGF-β, or VEGF-A primary antibody (1:300 dilution) at 4 °C and then incubated with secondary antibody. Sections were stained with diaminohydrazine and counterstained with hematoxylin and then dehydrated and cleared.

For ELISA assay, the extracted tissues were washed with prechilled PBS (0.01 M, pH = 7.4) to remove residual blood. After weighing, the tissue was cut into small pieces. For every 10 mg of muscle tissue, 100 μl of NP-40 lysis buffer containing phenylmethylsulfonyl fluoride (Beyotime) was added, and the mixture was homogenized with an ice electric homogenizer for 10 minutes. Then the tissue homogenate was centrifuged at 4 °C for 10 min. The supernatant was collected and stored at −80 °C. To measure the concentrations of growth factors in the supernatant of muscle tissue, the TNF-α, IL-1β, IL-6 and IL-10 rat ELISA kits (Abcam, UK) can be used according to the manufacturer's instructions.

## Statistical and reproducibility

At least three independent experiments of each type have been done and produced consistent results. All independent samples in this study represent the number of samples, tissues or animals in each group, with only one result obtained for each sample or animal. All analysis data were expressed as mean ± standard deviation (at least three or more independent sample). The statistical difference among different groups was accessed via two-sided one-way analysis of variance (ANOVA) methods. Image J, Origin 2018b and Excel were used for data analysis and plotting. Differences of $*p < 0.05$ denote significance, while differences of $**p < 0.01$ and $***p < 0.001$ denote high significance compared with other groups, ns denote no significance ($P \geq 0.05$).

## Ethics

Every experiment involving animals, human participants, or clinical samples have been carried out following a protocol approved by an ethical commission.

## Reporting summary

Further information on research design is available in the Nature Portfolio Reporting Summary linked to this article.

## Data availability

All data supporting the findings of this study are available within the article and its supplementary files. Any additional requests for information can be directed to, and will be fulfilled by, the corresponding authors. Source data are provided with this paper.

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

## Acknowledgements

We gratefully acknowledge the financial support by Scientific Research Innovation Capability Support Project for Young Faculty (ZYGXQNJS-KYCXNLZCXM-M5), National Natural Science Foundation of China (No. 82301331, No. 82330034, and No. 82271161), Key Technologies Research and Development Program (2024YFC2418303), DHU Distinguished Young Professor Program (LZA2023001), and the Fundamental Research Funds for the Central Universities (CUSF-DH-D-2025005).

## Author contributions

C.Y.H., S.K.Y., H.W., and L.P.L. guided the project. Z.Q.S. conceived the idea and designed the experiment. Z.Q.S. fabricated the BET-suture. Z.Q.S., Y.F.J., and H.S. performed the experiments and measurements. S.K.Y., and H.W. guided the biomolecular mechanism. C.Y.H., Y.G.L., Q.H.Z., and K.R.L. revised the manuscript. All authors analyzed the experimental data, drew the figures and prepared the manuscript. All authors discussed the results and reviewed the manuscript.

## Competing interests

The authors declare no competing interests.
