## [Transparent Peer Review file · Nature Communications]

A bioabsorbable body-coupling-electrotherapy suture

Corresponding Author: Professor Chengyi Hou

Version 0:

Reviewer comments:

Reviewer #1

(Remarks to the Author)

The authors presented Mo/PLGA/Ag NPs-based body-coupling-electrotherapy suture. BET-suture can generate electrical stimulation to the wound using environmental electromagnetic field. It is good idea to harvest meaningless energy, but it arouse user safety concern when the patient is exposed strong electric field in real world.

1. There are several body-coupling system to generate electrical signals for various purposes. What is the new finding of the manuscript?
2. BET-suture acts as electromagnetic receiver, but it cannot regulate its voltage output, waveform, etc. If the BET-suture expose high electromagnetic field, it generate excess voltage, which cannot guarantee user safety.
3. The authors argued the BET-suture can charge energy (Figure 3c). How it charge energy when they receive AC signals? Why existence of Ag NPs dramatically increase charge storage?
4. Figure S16 show antibacterial properties. The results look like Ag NPs kill bacteria due to its toxicity. It needs to provide electrical information of the BET group and to prove its capacitive antibacterial properties. In addition, voltage applied group using wave generator is required to confirm electrical antibacterial effect.
5. Applied electric field (5V/mm) is too high. It will be able to make electrolytic hydrolysis and burn at wound.
6. Number of sample for in vivo experiments are too low.

Reviewer #2

(Remarks to the Author)

The authors proposed a bioabsorbable body-coupling-electrotherapy suture for comprehensive wound regulation from the inflammation-remodeling stage, promoting healthy tissue healing. The conversion of body-coupled electromagnetic energy through the suture enabled electrically synergistic therapeutic capabilities and the electrical stimulation ability based on dielectric voltage difference promoted high expression of healing factors. After reviewing this article, I have a few major concerns with the developed self-driven ES suture and its energy generation approach before decision-making. Therefore a major revision is requested as follows:

Comments:

(1) As displayed in Figure S1c and Video S1, it seems that the body EM coupling necessitates the system ground in conjunction with the earth ground. This indicates that the system ground (from electronic appliances or measuring equipment- in this case, Oscilloscope) is always required for the flow of EM energy from the body and this could possibly be altered by the power supply AC noise induced from the ground pin of the measuring equipment itself and the human body. Considering this scenario it will create complexity to implement this method for wearable or implantable and or wound healing applications during outdoor moving conditions thereby restricting its self-powered functionality to lab tests.

(2) In the LED lighting setup, authors showcase the energy coupled between the human hand and the system ground to power LED which is a commonly observed phenomenon. Is it still possible to harvest EM energy to turn on LED solely due

to the polarization-induced electric field of the BET suture without the next terminal connected to the system ground? If this is not the case, its more complex self-powered strategy and supplying conventional electricity to stimulate wound healing would be much more efficient. Authors are advised to demonstrate LED lighting with the proposed BET suture instead of the wiring cables.

(3) The body-coupled EM energy-induced voltage waveforms are displayed. However, there is no detailed electrical analysis such as the amount of EM energy that can be harvested using the proposed BET suture. Authors are advised to supplement the additional experimental data such as output current and output power and/or energy output with distance variation.

(4) Detailed analysis is necessary to estimate the energy required to facilitate the electrotherapy for a comprehensive wound healing process and the overall power consumption using the traditional approaches.

(5) In Figure 3b, electric signals were supplied using a function generator to estimate the stored charges in a suture. How about the estimation of the stored charges in the suture solely due to the body-induced EM energy and what about the influence of an external EM field from the surrounding appliances on the stored charges in a suture?

(6) What is the rationale between the variation in continuously measured voltage results in an oscilloscope and the wound healing steps itself if in case, the self-degradation properties of the suture were neglected? Will the healing states of the wounded tissues affect the generated EM voltage?

(7) Authors stated that stitching only with the core layer of BET suture induced zero output voltage and the voltage induction was solely the effect of the body-induced EM polarizing the dielectric layer. On the contrary, the LED demonstration using the conducting wire without any dielectric layer was able to generate an electric voltage sufficient to operate the LED? How much is the conductivity of the core layer of BET suture? How does it affect the overall induced potential?

Reviewer #3

(Remarks to the Author)

Sun et al. developed a novel bioabsorbable body-coupling-electrotherapy suture, that also been characterized in various aspects regarding mechanical properties, antimicrobial efficiency, and wound healing. The authors explained the state of the art in the manuscript by representing several comprehensive results. The manuscript is well-organized and thoroughly discussed, and it has the potential to be published in Nature Communications after minor revisions.

1. In figure 3a, AgNPs was depicted as a surface charge of (+). Does the surface charge due to the capping agents or surface ligands that are attached on the AgNPs surface? In the materials section, it is mentioned that Ag NPs was purchased commercially but no information was provided further in terms of structure. The reviewer would like to see detailed information on the surface charge of the AgNPs, where zeta potential experiments can be considered as a tool to support the claim of positively charged AgNPs.

2. Could the authors provide the information for;
What is the difference between blue and grey bars in Fig.S3c? Could authors clarify it in the figure caption of Fig. S3?

3. Regarding degradation mechanism – from Fig.S5 it looks like a surface etching starts in day 10. In this scenario, the physiological environment will also start to degrade Mo starting from the day 10th, because the outer layer (PLGA) starts to dissolve at the 10th day.

Would this cause an altering performance of BET suture because of degradation of Mo?

4. Following the previous comment, a positive potential (oxidation) can accelerate the degradation of Mo by the formation of molybdenum oxides (ChemSusChem 2012, 5, 1146 – 1161). Did authors perform any degradation test on a Mo filament under applied voltage? Because this would directly affect the performance of the ES, especially after the beginning of PLGA degradation.

5. The authors mentioned clearly about the toxicity of the AgNPs. However, the surface of AgNPs usually functionalized with capping agents or ligands to form stable AgNPs dispersions.

Could these type of molecules (capping agents or ligands) affect the toxicity of AgNPs? I assume that after the AgNPs metabolization as in Fig. S5a the ligands will be released to the medium, and in what extent this molecules could have an impact on toxicity levels?

6. In the manuscript (line 136) the authors mentioned the synthetic skin. Could authors provide more information about the synthetic skin? - Was it provided by commercial source? (if so please provide the source in the material section), or was it fabricated in the lab? (if so, please provide the method for fabrication)

7. The drag force experiment was performed on synthetic skin. Did authors try any comparative study for drag force test between synthetic skin and mouse skin? How would the drag force change depending on the skin model?

8. Does the commercial silver nanoparticles (Ag NPs, 10-500 nm) in solution? If so, could the author briefly explain how did they calculated the weight percentage of AgNPs, which has been used in the fabrication of suture?

9. The reviewer is a little skeptical to refer AgNPs as a bioabsorbable material.

AgNPs forms Ag^+ ions (as it depicted Fig.S5a), which leads partial solubility of AgNPs in body. Further, Ag^+ can be filtered by the kidneys and excreted in urine. However, this does not mean that they are completely metabolize such as PLGA. Therefore, it would be nice to mention briefly in the article about the AgNPs, which is a not fully bioabsorbable material.

Reviewer #4

(Remarks to the Author)

At its core, this manuscript represents an integration of the authors' previous work published in Science ("Single body-coupled fiber enables chipless textile electronics") and Nature Communications ("A bioabsorbable mechanoelectric fiber as electrical stimulation suture"). However, the integration of these prior studies is not immediately apparent and demands scientific innovation—achievements that are indeed realized in this manuscript. For this reason, these reviewers find the work scientifically compelling and of potential interest to the readership of Nature Communications. Nevertheless, several major concerns must be addressed before the manuscript can be considered for publication.

1. Justification for Use of AgNPs

There is currently no sufficient justification for the choice of silver nanoparticles. Several questions remain unanswered:

- Why was silver selected over other biocompatible or biodegradable materials, such as Au, Mg, or Zn?
- Why this specific size and shape? The reported diameter range of 10–500 nm is broad. Does this variability impact conductivity, biodegradability, or biocompatibility?
- Were other nanoparticle forms such as nanowires (AgNWs) or nanostars considered?

The authors are encouraged to provide a rationale grounded in literature or experimental exploration of the influence of nanoparticle size, shape, and composition on system performance, which would significantly strengthen the conclusions.

2. Regulation of Maximum Current and Translational Considerations

The manuscript lacks discussion and data on how the system regulates maximum current. While it is understood that the device passively harvests EM energy, in real-world conditions, EM exposure can vary significantly. For example: Hospital environments with strong electromagnetic fields, Industrial settings, Proximity to high-voltage infrastructure (at safe distances). Could such conditions cause excessive current generation, potentially leading to overstimulation or malfunction? Given that this is the third paper from the group on this topic, it is time to begin considering and addressing such translational and safety concerns.

3. Improvement of Comparison Tables (Table S2 and Table S3)

Table S3 should include quantitative metrics such as wound healing time or the enhancement of healing through electrical stimulation compared to passive controls. Reference data from the authors' previous studies should be included to facilitate comparison. Additionally:

- Consistent terminology should be used across works. For instance, the therapeutic technique (electrical stimulation) is consistent, even if the delivery method differs.
- Table 2 lacks justification for material selection. Why choose Mo over alternatives like W or Mg? Why PLGA and not Ecoflex or PEG? A clearer rationale supported by quantitative data is needed.

4. Clarification of In Vivo Experimental Protocol

The in vivo experiments need a precise description:

- Line 646 indicates rats were euthanized at days 3 and 7, yet the experiment is described as running for 35 days.
- A clear schematic timeline of the experimental protocol should be added.
- Specifically, how were the crocodile clips attached to the animals? Were they secured in a way that guarantees stable and consistent attachment throughout the entire 35-day period?
- What ethical considerations support maintaining such a connection on a live animal for this duration? Is the clip directly attached to the sutures, or another anatomical site? Including detailed images or photographs of the connection method would greatly improve transparency and understanding of the experimental design. If the banana cables are connected directly to the animal's body, this reviewer recommends that the related data be excluded from the manuscript.

5. Terminology Inconsistency (ES-Suture vs BET-Suture)

The term "ES-suture" is confusing, as it was previously used to describe the mechanically activated suture. In this manuscript, it appears to refer to a BET-suture without AgNPs. Consistent and clearly defined terminology is essential to avoid confusion.

6. Definition of "Optimal Recovery Status" (Figure 5b, Line 413–414)

The claim that the BET-suture shows "optimal recovery status" needs a more precise and evidence-based explanation. What metrics define "optimal"? Histology? Functional recovery? Please elaborate clearly.

7. Sample Clarification (Figure S25)

The statement that $n = 3$ independent samples were used for mouse measurements requires clarification. Does this mean three different animals, or three repeated measurements from a single animal? The sample size and method of measurement should be clearly stated.

Minor Comments

8. The novelty of this work compared to the previous two papers is not clearly articulated. A concise summary should be added to the Introduction, explicitly stating: What was achieved in each prior paper, What limitations remained, How this work addresses those limitations. Specifically, the key limitation from the Science paper—insufficient current from EM energy harvesting—seems to be overcome in this work via AgNP-enhanced conductivity. If this is indeed the core novelty, it should be clearly emphasized in the abstract and intro.

9. The introduction of AgNPs in the Results section is abrupt. Given their biocompatibility profile, they should be introduced earlier. Since the data supports their safe use in this context, this should be highlighted as a novel achievement in both the abstract and intro.

10. Figure S2 is currently schematic. The authors are encouraged to include actual images or short videos of the fabrication

process. This will help the reader better understand the reproducibility and reliability of the method.

11. (Line 65) The sentence: "But scenarios that require dynamic conditional use limit its application to restricted area," is vague. What are these scenarios? Please provide concrete examples and references, or revise the sentence for clarity.

12. Figure 1b: Scale bar is not visible.

13. Figure 2a: The conductive path and AgNPs orientation are not clearly illustrated.

14. Figures 3 & 4: Staining images lack visibility and should be enhanced for clarity.

15. Figure 5a: A clearer image of the mouse, including visualization of suture placement and a timeline of the experimental procedure, would be helpful as discussed.

16. Consider including a comparative timeline in Figure 1 or visual summary of suture technologies—from antimicrobial chemical sutures to the current work with physical/electrical stimulation. This would help multidisciplinary readers position this work in the broader technological landscape, including existing limitations and the pathway to clinical translation.

Reviewer #5

(Remarks to the Author)

Reviewer #6

(Remarks to the Author)

Version 1:

Reviewer comments:

Reviewer #1

(Remarks to the Author)

The authors have revised the manuscript in response to the reviewers' comments. However, several important issues remain to be addressed:

1. Body-coupled or body-mediated electronic systems are already well-established technologies. Similarly, PLGA and Mo are widely used bioresorbable materials. The authors should clearly articulate the novelty of their work beyond these existing approaches.

2. In Figure R2, the reported peak voltage exceeds 15 V, which is unusually high for an implantable system. In particular, when an individual passes near high-voltage power lines (typically above 230 kV), there may be potential safety hazards for the user. The authors should address this concern.

3. In Figure R3b, it is not clear why the PLGA/Ag NP structure does not outperform the Mo-based structure. Further clarification and mechanistic explanation are required.

4. The claim that the charge storage and release capacity of Ag NPs contributes to antibacterial properties is not convincing. Since electrical stimulation alone does not effectively reduce bacterial populations, the antibacterial effect should be evaluated under conditions without electrostimulation using the BET. If antibacterial activity is observed, it is more likely attributable to the intrinsic toxicity of Ag NPs rather than their charge-storage capability.

5. Figure R5b is confusing. In Figure 5, the applied peak voltage of 5 V corresponds to approximately 9 V/mm of electric field (see Figure R5a). Did the authors calculate the electric field under the actual animal experimental conditions? Figure R5b suggests that an electric field exceeding 5 V/mm can cause cellular damage. Therefore, the safety of the system should be carefully analyzed and discussed.

Reviewer #2

(Remarks to the Author)

The authors have appropriately responded to my comments and revised the manuscript accordingly. Therefore, I can recommend the revised version of the manuscript for publication in this journal.

Reviewer #3

(Remarks to the Author)

I appreciated the effort the authors made to address all my comments by providing clear descriptions and additional

experiments. I am pleased to recommend the article for publication in Nature Communication.

Reviewer #4

(Remarks to the Author)

The authors have comprehensively addressed all of our concerns and provided experimental evidence to support their claims. The data presented are consistent and convincing. Therefore, we recommend acceptance of the paper at this stage, with no further revisions required.

Reviewer #5

(Remarks to the Author)

Reviewer #6

(Remarks to the Author)

Version 2:

Reviewer comments:

Reviewer #1

(Remarks to the Author)

The author addressed all my questions.

Responses to the Reviewers' Comments

We sincerely thank the reviewers for their careful and thorough review. We have revised our manuscript very carefully in the light of their suggestions and comments.

The following responses have been prepared to address all of the reviewers' comments in a point-by-point fashion. (Comments in black, responses in blue):

Response to Reviewer #1

General comment: The authors presented Mo/PLGA/Ag NPs-based body-coupling-electrotherapy suture. BET-suture can generate electrical stimulation to the wound using environmental electromagnetic field. It is good idea to harvest meaningless energy, but it arouse user safety concern when the patient is exposed strong E-field in real world.

Response: Thank you for the positive feedback. We have carefully revised the manuscript according to your comments. The replies to each of your concern are listed below. In comment 2, we systematically explained the safety concerns that may arise for users when exposed to strong E-fields.

1. There are several body-coupling system to generate electrical signals for various purposes. What is the new finding of the manuscript?

Response: Thank you for raising the key comment regarding the new finding. There are several studies that utilize body-coupled system for various applications. Li et al. (Nature Electronics 2021, 4, 530-538) designed an EM energy harvesting circuit and transmitted electricity through body coupling to power wearable electronic devices such as watches. Subsequently, Yong et al. (ACS Energy Letters 2023, 8, 2954-2961) used copper patches that couple EM energy to stimulate muscles and improve muscle fatigue. Similarly, Kang et al. (Advanced Materials 2024, 36, 2402491) employed a similar patch device to couple EM energy for skin wound healing. Most recently, our research group (Science 2024, 384, 74-81) designed a fiber that can utilize EM energy for human-environment interaction, achieving a chipless interaction mode.

However, all such energy interaction modes were realized through external media between the human body and the environment, without delving into the interaction form and application value of implantable electronic devices and electromagnetic energy (Figure R1).

In this work, the key novelty lies in pioneering a bioabsorbable, implantable fiber-based system, which for the first time proposes the in-vivo conversion mechanism of body-coupled EM energy. Specifically, it utilizes the energy storage properties of sheath layer with high dielectric performance to construct a potential difference for ES and capacitive

antibacterial capability at the wound site, thereby achieving comprehensive wound regulation from the inflammation-to-remodeling stage and promoting healthy tissue healing. **This wound repair capability has never been reported in previous studies on intelligent sutures (Table S3).** This represents a fundamental shift—from external energy coupling to internalized, therapeutic coupling—and opens new avenues for minimally invasive, self-powered medical devices.

a In vitro interaction (References)

b In vivo therapy (This work)

Figure R1. (a) EM energy reported in the references interacts with the human body and the environment through an external media. (b) This work proposes the interaction form between implantable electronic devices and EM energy and its therapeutic application in vivo.

● **Our revision to the manuscript:**

We added “We discussed the development of the body-coupled system in Note S1.” (see Introduction) in the revised manuscript and added “Note S1. BET-suture realizes body-coupled systems for implantable applications” in the revised supplementary materials.

Corresponding changes have been marked in red in the revised manuscript and supplementary materials.

2. BET-suture acts as electromagnetic receiver, but it cannot regulate its voltage output, waveform, etc. If the BET-suture expose high electromagnetic field, it generate excess voltage, which cannot guarantee user safety.

Response: Thank you for your concern for user safety. The voltage output and waveform can be adjusted by the environmental EM field. As shown in Figures R2a and b, we can achieve changes in the body-coupled energy and frequency by adjusting the strength and frequency of the environmental EM field. This allows for adaptation to the ES intensity and frequency required for different treatment areas.

In fact, as shown in Figure R2c (Figure S1b in the original manuscript), in places where humans frequently gather (offices, laboratories, outdoors, factories, etc.), the body-coupled EM energy is only at the microwatt (-20 dBm) level and does not pose a hazard to humans. Additionally, as shown in Figure R2d (Figure 2g in the original manuscript), the body-coupled voltage decreases rapidly with increasing distance. In special cases where exposed to high EM fields, maintaining a safe distance will prevent harm to users. For example, the national standard (DL 408-2016 Electrical Safety Work Procedures) specifies the relationship between voltage levels and safe distances to ensure personal safety during work operations.

We further illustrate that maintaining a safe distance does not pose a hazard to humans with the following formula (Note S3 in the original manuscript):

$$\overrightarrow{V_{incision}} = \overrightarrow{V_b} - \overrightarrow{V_d}$$

At a safe distance, the V_b generated by the body-coupled EM energy will remain within a safe range. The above formula indicates that the entire ES system is powered by body-coupled EM energy, and the V_d generated by the BET-suture is transmitted by V_b to create a potential difference ($V_{incision}$). Therefore, as long as environmental exposure remains within regulated levels (i.e., safe distances are maintained), the induced voltage and current cannot exceed safety limits.

Figure R2. Variation trends of body-coupled voltage (V_b) with changes in (a) strength and (b) frequency of the environmental EM field. (c) Power spectrum of ambient EM waves coupled by the human body and air in a laboratory environment. (d) V_p of $V_{incision}$ at different distances from the transmitter.

● **Our revision to the manuscript:**

We added Figure R2a and b as Figure S2a and c in the revised supplementary materials. In addition, we added “Even under complex E-fields, maintaining a safe distance would ensure personal safety (Figure S2a). Meanwhile, within the frequency range of biological electrical signals (10–90 Hz), it is possible to couple a stable voltage (Figure S2b).” (see part 2.1 Design and principle of bioabsorbable body-coupling-electrotherapy suture) to emphasize the safety of body-coupled voltage in the revised manuscript.

Corresponding changes have been marked in red in the revised manuscript and supplementary materials.

3. The authors argued the BET-suture can charge energy (Figure 3c). How it charge energy when they receive AC signals? Why existence of Ag NPs dramatically increase charge storage?

Response: Thank you for your careful review. Dielectric materials can continuously store and release energy in AC signals. Figure 3c showed the process of charge accumulation until saturation in the sheath layer of BET-suture under a 50 Hz AC signal. This process could be monitored using a Keithley 6514 electrometer (Keithley Instruments, USA), which accurately captured the transient charge accumulation.

The incorporation of Ag NPs significantly enhances the charge storage capacity of the sheath. This improvement arises from two key mechanisms: (1) the Ag NPs increase the relative permittivity of the sheath material (Figure R3a, corresponding to Figure S10b in the original manuscript), which allows more charge to be stored per unit electric field; and (2) the presence of Ag NPs facilitates faster charge polarization and relaxation dynamics (Figure R3b, corresponding to Figure S17d in the original manuscript), enabling more efficient charge storage and release within each AC cycle.

These enhancements collectively contribute to the superior capacitive antibacterial performance observed in the BET-suture.

Figure R3. (a) Variation curves of relative permittivity with frequency for sheath layers with different Ag NPs contents. (b) Real-time transferred charge densities of Mo, PLGA and PLGA/Ag NPs at a fixed AC.

4. Figure S16 show antibacterial properties. The results look like Ag NPs kill bacteria due to its toxicity. It needs to provide electrical information of the BET group and to prove its capacitive antibacterial properties. In addition, voltage applied group using wave generator is required to confirm electrical antibacterial effect.

Response: Thank you for your professional advices. We have clarified the electrical conditions and added further comparative analysis to support the capacitive antibacterial mechanism.

The BET group (with Ag NPs) and the BET-no-Ag NPs group were both subjected to a 50 Hz, 5 V_{rms} alternating electrical signal. A control group using the same BET-suture was not electrified. As shown in Figure R4 (Figure S18 in the original manuscript), the BET group exhibited significantly stronger antibacterial performance than the BET-no-Ag NPs group, despite both being exposed to the same electrical intensity. This indicates that the enhanced antibacterial effect is not due to ES, but rather attributed to the increased charge storage and release capacity of the Ag NP-containing dielectric sheath, thus supporting the capacitive antibacterial mechanism.

Furthermore, the results of the BET-no-Ag NPs group and the control group were

compared to assess whether electrical stimulation alone (within a wound-safe range) could effectively kill bacteria. The results showed that while 5 V_{rms} at 50 Hz is beneficial for promoting wound healing, it was insufficient to exert direct bactericidal effects. This aligns with previous reports that bacterial membranes typically possess higher voltage resistance than mammalian cells.

In summary, the results indicated that within a physiologically safe electrical range, the incorporation of Ag NPs enhances dielectric properties and enables capacitive charge-mediated antibacterial activity, effectively preventing early-stage wound infections.

Figure R4. Capacitive antimicrobial capacity of BET-suture. (a) Photographs of the proliferative state of *S. aureus* and *E. coli* on LB plates after treatment in each group. Scale bar: 1 cm. Cell counting statistics for (b) *S. aureus* and (c) *E. coli* on LB plates. n = 3 independent samples. All statistical analyses were performed by one-way ANOVA, data represent mean ± standard deviation, **p < 0.01, *p < 0.05, ns indicates not significant.

- **Our revision to the manuscript:**

We added “The electrical parameters for the BET group and the BET no-Ag NPs group were 50 Hz and 5 V_{rms}.” (see part 4.6 Antibacterial activity) to describe the electrical parameters and “This indicated that the enhanced antibacterial effect was not due to ES, but rather attributed to the increased charge storage and release capacity of the Ag NP-containing dielectric sheath.” (see part 2.3 Charge-enhanced antimicrobial effect based on capacitive properties) in the revised

manuscript.

Corresponding changes have been marked in red in the revised manuscript.

5. Applied E-field (5V/mm) is too high. It will be able to make electrolytic hydrolysis and burn at wound.

Response: Thank you for your professional advices. We fully agree that excessive E-field strength may lead to undesired effects such as electrolytic hydrolysis or thermal damage at wound sites. In our study, we have systematically investigated the safety thresholds of E-field intensity through both simulation and experimental validation.

Computational analysis – COMSOL simulations were conducted to map the local E-field distribution under different applied voltages and voltage-E-field relationship curves were fitted (Figure R5a, corresponding to Figure S21 in the original manuscript). The method was utilized to calculate the ES intensity in the in vitro cell experiments to determine the threshold value. Similar methodologies and thresholds have been reported in previous studies (Nature Electronics 2024, 7, 299–312; Nature Biomedical Engineering 2025, DOI: 10.1038/s41551-025-01440-6; PNAS 2021, 118, e2100772118).

Experimental validation – We applied different stimulation voltages based on the fitted curves. As shown in Figure R5b (Figure S23 in the original manuscript), under stimulation conditions with a maximum E-field strength below 5 V/mm, cell migration and proliferation were effectively enhanced without any observable damage. In contrast, stimulation conditions with a minimum E-field strength exceeding 5 V/mm caused a degree of cellular damage, including inhibition of migration and reduced viability (Figure R5c, corresponding to Figure 4c in the original manuscript).

These results collectively confirm that our applied E-field is within a biologically safe range and does not induce electrolytic or thermal damage during stimulation.

Figure R5. (a) COMSOL simulation of the E-field distribution of BET-suture in cell petri dish. (b) Effect of E-field strength on cell proliferation provided by BET-suture. Scar bar: 500 μm . (c) Microscopic images after 24 h of the scratch healing assay and Ki67 fluorescence staining images after 72h of the cell proliferation assay. Scar bar: 80 μm .

6. Number of sample for in vivo experiments are too low.

Response: We added two new sets of data to each of the four animal experiments. The results of the new animal experiments were consistent with the results of the previous experiments. These results included wound healing rate, CD31 and α -SMA fluorescence staining, CD3 and CD20 fluorescence staining, growth factor immunohistochemistry, and inflammatory cytokine ELISA experiments.

● **Our revision to the manuscript:**

The relevant data are supplemented in Figure 5e, Figure 5g-i, Figure S29b and c and Figure S30b and c in the revised manuscript and supplementary materials.

Corresponding changes have been marked in red in the revised manuscript and supplementary materials.

Thank you again for your valuable comments and suggestions.

Response to Reviewer #2

General comment: The authors proposed a bioabsorbable body-coupling-electrotherapy suture for comprehensive wound regulation from the inflammation-remodeling stage, promoting healthy tissue healing. The conversion of body-coupled electromagnetic energy through the suture enabled electrically synergistic therapeutic capabilities and the electrical stimulation ability based on dielectric voltage difference promoted high expression of healing factors. After reviewing this article, I have a few major concerns with the developed self-driven ES suture and its energy generation approach before decision-making. Therefore a major revision is requested as follows:

Response: Thank you for your review and we appreciate your feedback. We have carefully revised the manuscript according to your comments. The replies to each of your concern are listed below.

1. As displayed in Figure S1c and Video S1, it seems that the body EM coupling necessitates the system ground in conjunction with the earth ground. This indicates that the system ground (from electronic appliances or measuring equipment- in this case, Oscilloscope) is always required for the flow of EM energy from the body and this could possibly be altered by the power supply AC noise induced from the ground pin of the measuring equipment itself and the human body. Considering this scenario it will create complexity to implement this method for wearable or implantable and or wound healing applications during outdoor moving conditions thereby restricting its self-powered functionality to lab tests.

Response: Thank you for your professional advices. The principle of body-coupled EM energy harvesting is based on the fact that the human body has a high relative permittivity and conductivity ($\epsilon \approx 78$ and $\sigma \approx 0.6$ S/m, respectively), which generates quasi-static polarization in an EM field, coupling the EM field and forming an induced voltage (Science 2024, 384, 74-81). This results in a displacement current path between the body and surrounding objects at different potentials, which can be the building floor, nearby conductive structures, or even the environment itself. In outdoor or mobile conditions, this parasitic coupling still exists, forming a closed loop via capacitive displacement currents rather than requiring a direct lab ground (Figure R6a).

In Figure S1c and Movie S1, we need to ground one electrode of the LED to form a circuit and light up the LEDs. As shown in Figure R6a and b, the body couples ambient EM energy and forms a parasitic capacitive path to environmental ground (e.g., earth ground). When the body contacts the LED circuit, an ohmic contact is made with the electrode, delivering the coupled energy to the load for lighting up the LEDs.

In addition, we measured the voltage fluctuations of the body-coupled EM energy in different states under normal surroundings. As shown in Figure R6c and Movie R1, as long as the strength of the EM field in space remains constant, the potential will remain within a constant range regardless of how the human body moves.

Therefore, these results indicated that the operation does not depend on laboratory earth ground, and the principle remains valid for wearable, implantable, and wound-healing applications in mobile conditions.

Figure R6. (a) Schematic diagram of power transmission for body-coupled EM energy-driven LEDs. (b) Body-coupled EM energy directly lighting up three LEDs connected in series. (c) Real-time voltage output of body-coupled EM energy in different movement states, including standing, walking, running, and jumping.

● **Our revision to the manuscript:**

We added Figure R6c as Figure S2d and Movie R1 as Movie S2 in the revised supplementary materials. In addition, we added “**The coupled voltage is also unaffected by the individual's movement state (Figure S2d, Movie S2).**” (see part 2.1 Design and principle of bioabsorbable body-coupling-electrotherapy suture) in the revised manuscript.

Corresponding changes have been marked in red in the revised manuscript and supplementary materials.

2. In the LED lighting setup, authors showcase the energy coupled between the human hand and the system ground to power LED which is a commonly observed phenomenon. Is it still possible to harvest EM energy to turn on LED solely due to the polarization-induced E-field of

the BET suture without the next terminal connected to the system ground? If this is not the case, its more complex self-powered strategy and supplying conventional electricity to stimulate wound healing would be much more efficient. Authors are advised to demonstrate LED lighting with the proposed BET suture instead of the wiring cables.

Response: Thank you for your careful review. As mentioned in comment 1, the LEDs were lit due to body-coupled EM energy, not BET-suture. As shown in Figure R7a, when the return terminal of the LEDs circuit is left floating, the LEDs cannot be lit due to the absence of a closed loop, which is a fundamental requirement for current flow.

In response to your suggestion, we replaced conventional wiring with the BET-suture (Figure R7b), and observed no significant change in LED brightness. This is because the core layer of BET-suture functions the same as a conductive wire, serving solely to transmit body-coupled EM energy. Movie S1 primarily aimed to demonstrate that the human body, as a large dielectric, could effectively couple EM energy from the environment, providing an energy source for ES.

Figure R7. (a) Comparison of physical diagrams of grounded and ungrounded circuits driving LEDs to illuminate. (b) Physical diagrams and light intensity spectra of LEDs illuminated with and without BET-suture wiring.

3. The body-coupled EM energy-induced voltage waveforms are displayed. However, there is no detailed electrical analysis such as the amount of EM energy that can be harvested using the proposed BET suture. Authors are advised to supplement the additional experimental data such as output current and output power and/or energy output with distance variation.

Response: Thank you for your professional advices. In our system, the BET-suture does not independently harvest EM energy, but functions as a bioelectronic interface to transmit and convert body-coupled EM energy into a localized E-field at the wound site. As shown in Figure R8a, the EM energy is harvested via body coupling. The function of BET-suture is to convert

body-coupled EM energy in vivo and generate a potential difference at the wound site, thereby constructing an endogenous E-field to stimulate tissue repair. As shown in Figure R8b (Figure 2c in the original manuscript), we can measure the body-coupled voltage V_b and the induced voltage V_d in BET-suture, and calculate the potential difference $V_{incision}$ at the wound site through the following vector relationship:

$$\overrightarrow{V_{incision}} = \overrightarrow{V_b} - \overrightarrow{V_d}$$

We have provided a detailed analysis of the generation of potential differences in Note S3.

In addition, as shown in Figure R8c (Figure S1b in the original manuscript), the power of body-coupled EM energy can reach more than $10 \mu\text{W}$, which is 30-40 dBm higher than the power of air-coupled EM energy. Figure R8d (Figure 2g in the original manuscript) showed the relationship between energy output and distance. It could be seen that as the distance increased, the loss of EM waves in space transmission increased, resulting in a decrease in $V_{incision}$. However, it could still be maintained above the volt level to provide effective ES.

These results confirm that the system can achieve microwatt-level power harvesting and in situ voltage generation without external wiring, supporting its feasibility for self-powered biomedical applications.

Figure R8. (a) Comparison of BET-suture-coupled EM energy voltage output and body-coupled EM energy voltage output. (b) Real-time measurements of $V_{incision}$, V_b , and V_d during wireless ES. V_b , open-circuit voltage of body-coupled EM energy; V_d , open-circuit voltage of BET-suture's core layer. (c) Power spectrum of ambient EM waves coupled by the human body and air in a laboratory environment. (d) V_p of $V_{incision}$ at different distances from the transmitter.

4. Detailed analysis is necessary to estimate the energy required to facilitate the electrotherapy for a comprehensive wound healing process and the overall power consumption using the traditional approaches.

Response: Thank you for your professional advices. We have obtained the minimum threshold for inducing cell proliferation and migration behavior by using a Ca^{2+} fluorescent probe. As shown in Figure R9a and b (Figure 4d and e in the original manuscript), there is a minimum effective E-field strength threshold of 0.75 V/mm required for BET-suture to function. Subsequently, based on the E-field strength simulation of wounds (Figure R9c and d, corresponding to Figure 2g and Figure S14a in the original manuscript), the minimum voltage and current required for effective ES are 1.81 V and 1.12 μA , respectively. It is calculated that the minimum ES power of 2.03 μW .

We have added the power consumption of various ES devices to Table S2. Compared to ES devices powered by external power sources, primary cells, NFC and ultrasound, the BET-suture has lower electrotherapy power consumption.

While force-driven systems such as piezoelectric and triboelectric ES devices also operate at low power levels, they rely on external mechanical stimuli (e.g., body motion), which can be inconsistent and difficult to regulate. In contrast, the BET-suture allows remote control of stimulation intensity by tuning ambient EM field strength or adjusting the relative distance between the body and EM source, providing a more controllable and stable stimulation environment.

These findings support the potential of the BET-suture as a low-power, wireless alternative for sustained electrotherapy in wound healing applications.

Figure R9. (a) Representative fluorescence images of activated Ca^{2+} fluorescent probe in fibroblasts after ES action with applied voltages ranging from 0 to 140 mV. Scale bar: 30 μm .

(b) Plot of relative fluorescence intensity of intracellular Ca^{2+} probe against the applied ES voltage. $n = 3$ independent samples. (c) V_p of V_{incision} at different distances from the transmitter. (d) The maximum, minimum and median values were taken from the E-field results of the COMSOL simulation and the average E-field strength was calculated.

Table S2. Driving means and principles of various types of electrical stimulation.

Source of ES	Device shape and materials	ES power consumption	ES formation process	Ref.
External power supply	Patch: Mg, Mo, PLGA, PA	$>10^2$ mW	The electrodes are connected to an external power source to create an E-field	[5]
	Patch: MoS_2 , Al_2O_3 , HfO_2			[6]
Primary cell	Bandage: Mo, Mg, PCL, chitosan	$10 \mu\text{W}-10^2$ mW	Asymmetric electrodes encapsulated in a gel form a primary battery applying an E-field	[7]
	Patch: PVA/PBS, Mg, PAM-MXene-SF			[8]
	Patch: ecoflex, SMA grids, PTFE electret		Oppositely charged electret forms an E-field	[9]
NFC-driven	Bandage: flexible circuit boards	$1 \mu\text{W}-2$ mW	Using NFC to power electrodes on a circuit board to create an E-field	[10]
Ultrasound-driven	Patch: PHBV/PEG, Mg	$10^2 \mu\text{W}-10$ mW	Ultrasound-driven mechanoelectric conversion to form an E-field	[11]
	Patch: PCL-r-PU, PEDOT:PSS, PAV, Mo			[12]
Force-driven	Patch: Mg, PLGA	10 nW- $10^2 \mu\text{W}$	Force triggers piezoelectric or triboelectric effects to produce an E-field	[13]
	Patch: collagen, PLLA			[14]
	Suture: Mg, PLGA, PCL			[1]
Self-driven wireless	Suture: Mo, PLGA, Ag NPs	10 nW- $10^2 \mu\text{W}$	Body-coupled ambient electromagnetic energy drives suture to generate an E-field	This work

- **Our revision to the manuscript:**

We added “The first two types of devices consume more than $10^2 \mu\text{W}$ of power during therapy, which is significantly higher than force-driven ES devices and our BET-suture. But

force-driven ES devices, due to their mechanoelectric conversion mechanism, offer an almost arbitrary and uncontrollable stimulation intensity, with the potential to result in suboptimal therapeutic outcomes.” (see part 2.4 Threshold of electrical stimulation intensity and cell signaling pathways) in the revised manuscript.

Corresponding changes have been marked in red in the revised manuscript and supplementary materials.

5. In Figure 3b, electric signals were supplied using a function generator to estimate the stored charges in a suture. How about the estimation of the stored charges in the suture solely due to the body-induced EM energy and what about the influence of an external EM field from the surrounding appliances on the stored charges in a suture?

Response: Thank you for your insightful comment. In our study, the stored charge in the BET-suture during therapy is primarily generated via body-mediated EM coupling (Figure R10a, corresponding to Figure 1a in the original manuscript): Ambient EM fields polarize the human body ($\epsilon \approx 78$, $\sigma \approx 0.6$ S/m), which then transfers displacement currents to the BET-suture’s internal electrode. The PLGA/Ag NPs dielectric sheath enhances this process by storing the induced charge and creating a potential difference with surrounding tissue, enabling localized electrotherapy.

To quantify the charge contribution, we compared two conditions (Figure R10b):

Without body coupling: BET-suture placed in free space near EM sources \rightarrow transferred charge ≈ 0.02 μ C. With body coupling: human body in contact with BET-suture under same EM field \rightarrow transferred charge ≈ 0.15 μ C. This ~ 7.5 -fold increase confirms that the majority (>85 %) of stored charge originates from body-coupled EM energy, not from direct exposure of the suture to ambient EM fields.

Regarding the influence of surrounding electrical appliances: The BET-suture is not directly exposed to or responsive to environmental EM fields in free space. Instead, it relies on induced voltage through body-mediated polarization, meaning that fluctuations in external EM noise (such as nearby electronics) have minimal direct influence on the suture. In essence, the body acts as a mediator and filter, reducing susceptibility to environmental EM interference and enabling stable charge accumulation for localized electrotherapy.

Therefore, in Figure 3b, we used this combination device to demonstrate that the PLGA/Ag NPs dielectric sheath layer has higher charge storage capacity to justify the capacitive antibacterial capabilities of BET-suture.

Figure R10. (a) BET-suture senses and stores EM energy from the body-coupled environment, generating an E-field at the wound site to perform reparative functions. (b) The amount of charge transferred by body-coupled EM energy in BET-suture.

6. What is the rationale between the variation in continuously measured voltage results in an oscilloscope and the wound healing steps itself if in case, the self-degradation properties of the suture were neglected? Will the healing states of the wounded tissues affect the generated EM voltage?

Response: Continuous measurement of voltage changes was performed to demonstrate that BET-suture can provide stable and effective ES during the wound healing period.

The healing status of injured tissue does not affect the coupling voltage. The coupling voltage is determined solely by the dielectric properties of the biological body itself, such as volume, mass, and other factors. As shown in Figure R11a, it was found that the voltage generated using body-coupled EM energy was higher compared to rats. Furthermore, we stitched BET-suture into normal muscle, inflamed muscle, and fibrotic muscle, respectively, and measured the corresponding electrical signals (Figure R11b). The results showed that V_b , V_d , and $V_{incision}$ remained consistent across the three groups, further proving that the state of tissue healing does not affect the coupling voltage.

These findings demonstrate that the BET-suture provides a reliable and consistent source of in situ ES, regardless of wound healing phase or tissue condition, ensuring therapeutic continuity throughout the healing process.

Figure R11. (a) The voltage of human body-coupled and rat body-coupled EM energy. (b) Real-time measurements of $V_{incision}$, V_b , and V_d in different states of muscle tissue during wireless ES.

- **Our revision to the manuscript:**

We added Figure R11b as Figure S27d in the revised supplementary materials. In addition, we added “Furthermore, even when the wound was in an inflammatory or fibrotic state, it did not affect the ES performance of the BET sutures (Figure S27d).” (see part 2.5 Evaluation of incision healing effect in vivo) in the revised manuscript.

Corresponding changes have been marked in red in the revised manuscript and supplementary materials.

7. Authors stated that stitching only with the core layer of BET suture induced zero output voltage and the voltage induction was solely the effect of the body-induced EM polarizing the dielectric layer. On the contrary, the LED demonstration using the conducting wire without any dielectric layer was able to generate an electric voltage sufficient to operate the LED? How much is the conductivity of the core layer of BET suture? How does it affect the overall induced potential?

Response: As mentioned in Comment 1, LEDs are driven by body-coupled EM energy and are unrelated to BET-suture (Figure R12). This is to demonstrate that the human body can couple sufficient EM energy from the environment for subsequent ES by BET-suture.

In contrast, when discussing implanted applications, the mechanism is fundamentally different. Simply implanting the conductive core layer alone (without the dielectric sheath) results in no significant induced potential across the wound site, because the system lacks the necessary dielectric structure to accumulate and separate charges. As such, although the Mo filament core (with a conductivity of 1.89×10^5 S/cm) serves as an effective conductor to transport the body-induced EM energy, it cannot by itself generate a local electric field suitable for electrotherapy.

The energy storage properties of the dielectric sheath of the BET-suture are essential for capacitive charge accumulation, enabling the formation of a potential difference between the suture and surrounding tissues. This dielectric-induced polarization is what gives rise to the localized ES. Thus, while the conductive core determines how efficiently energy is transmitted, it is the dielectric layer that governs how much potential is induced and delivered for therapeutic stimulation. A detailed analysis of ES generation and calculation is presented in Note S3.

Figure R12. (a) Schematic diagram of power transmission for body-coupled EM energy-driven LEDs. (b) Body-coupled EM energy directly lighting up three LEDs connected in series.

Thank you again for your valuable comments and suggestions.

Response to Reviewer #3

General comment: Sun et al. developed a novel bioabsorbable body-coupling-electrotherapy suture, that also been characterized in various aspects regarding mechanical properties, antimicrobial efficiency, and wound healing. The authors explained the state of the art in the manuscript by representing several comprehensive results. The manuscript is well-organized and thoroughly discussed, and it has the potential to be published in Nature Communications after minor revisions.

Response: Thank you for positive feedback. We have carefully revised the manuscript according to your comments. The replies to each of your concern are listed below.

1. In figure 3a, Ag NPs was depicted as a surface charge of (+). Does the surface charge due to the capping agents or surface ligands that are attached on the Ag NPs surface? In the materials section, it is mentioned that Ag NPs was purchased commercially but no information was provided further in terms of structure.

The reviewer would like to see detailed information on the surface charge of the Ag NPs, where zeta potential experiments can be considered as a tool to support the claim of positively charged Ag NPs.

Response: Thank you for your professional advices. The purchased Ag NPs were in powder form, without any capping agents or surface ligands, and had a purity of $\geq 99.9\%$.

In Figure 3a, Ag NPs were depicted as Ag^+ since a certain amount of Ag^+ was involved in the antibacterial process. This was demonstrated in the animal experiments (Figure R13a, corresponding to Figure 5i in the original manuscript). On the third day, the antibacterial sutures loaded with Ag NPs showed a significant improvement in inflammatory infiltration compared to conventional absorbable sutures. This is due to the anti-inflammatory effect of Ag^+ release, and we have discussed the reasons for the inflammatory differences in detail in Note S6. Therefore, to more clearly illustrate the antibacterial mechanism of BET-suture, we have described the potential antibacterial effect of a small amount of Ag^+ in Figure 3a.

In addition, we measured the Zeta potential and particle size distribution of Ag NPs (Figure R13b and c). The Ag NPs were dispersed in deionized water and measured the Zeta potential and average particle size to be 9.44 mV and 199.27 nm, respectively. This indicated that the surface of Ag NPs carried a certain amount of Ag^+ , giving it a positive charge.

These findings support that the Ag NPs used in our system possess a positively charged surface under physiological conditions, which contributes to their antibacterial and anti-inflammatory functions in the BET-suture.

Figure R13. (a) Cytokines (TNF- α , IL-1 β , IL-6, IL-10) expression levels of the wound tissues after 3 days of stitching. (b) Zeta potential and (c) particle size distribution of Ag NPs.

- **Our revision to the manuscript:**

We added Figure R13b and c as Figure S11a and b in the revised supplementary materials. In addition, we added “In addition, we have provided relevant parameters and properties of Ag NPs (Figure S11), and discussed in Note S5 that the effects of different sizes of Ag NPs and silver nanowires on the dielectric properties of the sheath layer, as well as the advantages of selecting Ag NPs.” (see part 2.2 Mechanism of electrical stimulation based on dielectric voltage difference) and the measurement method “The Ag NPs were dispersed in a deionized water/ethanol solution, and the Zeta potential and particle size distribution of the Ag NPs were measured using dynamic light scattering (DLS, Malvern Zetasizer Nano ZS90, UK).” (see part 4.3 Materials characterization) in the revised manuscript, and added “Note S5. Size selection and advantages of Ag NPs” in the revised supplementary materials.

Corresponding changes have been marked in red in the revised manuscript and supplementary materials.

2. Could the authors provide the information for;

What is the difference between blue and grey bars in Fig.S3c? Could authors clarify it in the figure caption of Fig. S3?

Response: Thank you for your careful review. The blue and grey bars in Figure S3c represent the tensile strength and diameter of BET-suture twisted from 5, 8, and 11 strands of primary fibers, respectively. In the revised manuscript, Figure S3c has been renumbered as Figure S4c. We have provided detailed explanations in the revised supporting information for Figure S4c.

- **Our revision to the manuscript:**

We added detailed explanations “(d) Breaking tensile force (grey) and diameter (blue) of BET-suture twisted from 5, 8, and 11 strands of primary fibers, respectively. n = 3 independent samples. Scale bar: 200 μ m.” to the figure caption of Figure S4 in the revised supplementary materials.

Corresponding changes have been marked in red in the revised manuscript and supplementary materials.

3. Regarding degradation mechanism – from Fig.S5 it looks like a surface etching starts in day 10. In this scenario, the physiological environment will also start to degrade Mo starting from the day 10th, because the outer layer (PLGA) starts to dissolve at the 10th day.

Would this cause an altering performance of BET suture because of degradation of Mo?

Response: The degradation process of BET-suture proceeded from the sheath layer (PLGA/Ag NPs) to the core layer (Mo). Once the sheath layer was completely degraded, BET-suture no longer possessed ES capability. This was due to the energy storage (capacitance) properties of the dielectric sheath layer, which facilitated the formation of a potential difference between the electrode and the surrounding tissue, thereby inducing ES at the wound site. As shown in Figures R14a and b (Figures 5d and S27b in the original manuscript), BET-suture was able to maintain stable ES performance during the recovery period (7 days). After 10 days, the ES performance began to decline, with the fastest decline occurring between 14 and 21 days. This is because the sheath layer was rapidly degraded during this period (Figure R14c, corresponding to Figure S6c in the original manuscript), causing the BET-suture to lose its ES capability.

Therefore, variations in the ES performance of BET-suture are primarily influenced by the degradation of the dielectric sheath layer. We have provided a detailed analysis of the trends in electrical performance changes in Note S7.

Figure R14. (a) Real-time signals of $V_{injection}$ continuously monitored over 7 days of stitching. (b) Variation of V_p at different times of implantation. (c) Variations in the surface morphology of BET-suture during various stages of the degradation cycle. No visible BET-suture was seen in the degradation solution after 11 weeks. Scale bars: 10 μm for SEM images, 2 μm for SEM

insets, 10 μ m for microscope images, 3 cm for physical images.

4. Following the previous comment, a positive potential (oxidation) can accelerate the degradation of Mo by the formation of molybdenum oxides (ChemSusChem 2012, 5, 1146 – 1161). Did authors perform any degradation test on a Mo filament under applied voltage? Because this would directly affect the performance of the ES, especially after the beginning of PLGA degradation.

Response: Thank you for your professional advices. We further conducted degradation tests on Mo filaments under applied voltage conditions. As shown in Figure R15, Mo filaments were subjected to a simulated stimulation protocol (50 Hz, 5 V_{rms}, 1 hour every 12 hours) in a physiological solution for 3 days. The diameter of the non-electrified Mo filament remained at 84.02 μ m, whereas the electrified filament decreased to 65.1 μ m, and exhibited a roughened surface morphology—indicating accelerated surface corrosion under an applied voltage. This observation is consistent with prior studies on voltage-assisted oxidation of Mo (ChemSusChem 2012, 5, 1146–1161).

However, it is important to note that this accelerated degradation does not impair ES performance in practice. As discussed in Comment 3, the generation of ES is governed by the dielectric sheath layer, which provides capacitive property and the resultant E-field at the wound site. Once this dielectric sheath degrades (typically after day 10–14), the system naturally ceases its ES function, regardless of Mo filament integrity.

In other words, although the Mo core may begin to oxidize once exposed, this occurs after the active ES window, and thus has minimal impact on the therapeutic efficacy of BET-suture. Moreover, the controlled degradation of Mo contributes to the transient and bioresorbable nature of the system, aligning with the design goals for biodegradable bioelectronics.

Figure R15. (a) Photos of Mo filaments degrading in PBS solution (pH 7.4, 37 °C) under non-electrified (left) and electrified (right) conditions. (b) SEM images of Mo filaments after degradation for 3 days under non-electrified (left) and electrified (right) conditions.

5. The authors mentioned clearly about the toxicity of the Ag NPs. However, the surface of Ag NPs usually functionalized with capping agents or ligands to form stable Ag NPs dispersions. Could these type of molecules (capping agents or ligands) affect the toxicity of Ag NPs? I assume that after the Ag NPs metabolization as in Fig. S5a the ligands will be released to the medium, and in what extent this molecules could have an impact on toxicity levels?

Response: Thank you for your professional advices. The Ag NPs used in this study were purchased as unmodified powder nanoparticles with $\geq 99.9\%$ purity and no added capping agents or stabilizing ligands.

In our case, the Ag NPs were directly embedded in the PLGA sheath, and no exogenous stabilizing molecules were introduced. To ensure systemic biosafety, we performed histological analyses. After 28 days of BET-suture implantation, we collected tissues from vital organs (including lung, liver, spleen, kidney, and heart) and performed histological examination by H&E staining (Figure R16). The results showed that, compared to normal rat organs, the rat organs stitched with BET-suture did not exhibit signs of pathological inflammation or systemic immune responses, such as abnormal lymphocyte infiltration in vital organs, thereby ruling out the risk of functional impairment and organic lesions.

This indicated that the low amount of Ag NPs added in the BET-suture exhibited good biocompatibility and low toxicity, while achieving capacitive antibacterial capability. This addressed the biological hazards associated with excessive addition of Ag NPs as antibacterial agents in previous studies, providing experimental evidence for the safe design of silver-based nanocomposites.

Figure R16. H&E staining of vital organs (lung, liver, spleen, kidneys and heart) in normal rats and BET-suture rats after 28 days of stitching. Scar bar: 300 μm .

● **Our revision to the manuscript:**

We added Figure R16 as Figure S28 in the revised supplementary materials. In addition, we added “Subsequently, we collected tissues from vital organs (including lung, liver, spleen,

kidney, and heart) and performed histological examination by H&E staining (Figure S28). The results showed that, compared to normal rat organs, the rat organs stitched with BET-suture did not exhibit signs of pathological inflammation or systemic immune responses, such as abnormal lymphocyte infiltration in vital organs, thereby ruling out the risk of functional impairment and organic lesions.” (see part 2.5 Evaluation of incision healing effect in vivo) in the revised manuscript.

Corresponding changes have been marked in red in the revised manuscript and supplementary materials.

6. In the manuscript (line 136) the authors mentioned the synthetic skin. Could authors provide more information about the synthetic skin? - Was it provided by commercial source? (if so please provide the source in the material section), or was it fabricated in the lab? (if so, please provide the method for fabrication)

Response: Thank you for your careful review. The synthetic skin was purchased from Guangzhou Huijin Science and Education Model Co., Ltd. The main material is food-grade silicone, which realistically simulates the skin layer, fat layer, and muscle layer. We have added information about the source of the synthetic skin in the materials section.

- **Our revision to the manuscript:**

We added the source of artificial skin “The synthetic skin (food-grade silicone) was purchased from Guangzhou Huijin Science and Education Model Co., Ltd.” (see part 4.1 Materials) in the revised manuscript.

Corresponding changes have been marked in red in the revised manuscript.

7. The drag force experiment was performed on synthetic skin. Did authors try any comparative study for drag force test between synthetic skin and mouse skin? How would the drag force change depending on the skin model?

Response: Thank you for your professional advices. To assess the applicability of drag force data across tissue models, we conducted comparative drag force tests using synthetic skin, rat skin, and rat muscle tissues.

As shown in Figure R17, there are significant differences in drag force across different tissue models. For instance, muscle tissue, which has a higher density than skin, exhibits higher drag force. Importantly, the drag force values of BET-suture across all tissue models were found to be comparable to commercial medical sutures, and remained within the clinically acceptable range for safe suturing. This balance is essential, because excessively high drag force increases the risk of tissue trauma and tearing; Insufficient drag force can lead to inadequate wound

closure or suture slippage.

Therefore, BET-suture demonstrates appropriate drag force performance across different biological tissues, confirming its potential for practical surgical applications beyond synthetic skin testing.

Figure R17. (a) Photos of measuring the drag force of sutures in different tissue models, including synthetic skin, rat skin and rat muscle. (b) Comparison of drag forces of different medical sutures in different tissue models.

● **Our revision to the manuscript:**

We used Figure R17b to replace Figure S4f in the revised supplementary materials. In addition, we added “Moreover, we measured the tissue drag force per unit circumference when the suture was pulled through different tissue models (Figure S4f). BET-suture showed a traction force range of 3.91-4.66 N/cm, which is in the range between that of medical nylon sutures (5.68 N/cm) and silk sutures (2.89 N/cm).” (see part 2.1 Design and principle of bioabsorbable body-coupling-electrotherapy suture) and added the test method “The tissue drag force per unit circumference was measured using a tensile tester (Sanliang, SMF-50, Japan) when the suture was pulled through synthetic skin, rat skin, and rat muscle.” (see part 4.3 Materials characterization) in the revised manuscript.

Corresponding changes have been marked in red in the revised manuscript and supplementary materials.

8. Does the commercial silver nanoparticles (Ag NPs, 10-500 nm) in solution? If so, could the author briefly explain how did they calculated the weight percentage of AgNPs, which has been

used in the fabrication of suture?

Response: The commercial Ag NPs were in powder form. The Ag NPs are uniformly mixed with the PLGA solution to form the spinning solution for the sheath layer. The quantity of Ag NPs was determined based on the mass of PLGA, with the Ag NPs content expressed as a percentage of PLGA mass in the spinning solution. It was calculated by the following equation:

$$Wt\% = \frac{W_{Ag}}{W_{PLGA}} \times 100\%$$

Where W_{Ag} is the mass of Ag NPs in the spinning solution and W_{PLGA} is the mass of PLGA in the spinning solution. The calculated Wt% is the mass percentage of Ag NPs.

9. The reviewer is a little skeptical to refer Ag NPs as a bioabsorbable material.

Ag NPs forms Ag^+ ions (as it depicted Fig.S5a), which leads partial solubility of Ag NPs in body. Further, Ag^+ can be filtered by the kidneys and excreted in urine. However, this does not mean that they are completely metabolize such as PLGA. Therefore, it would be nice to mention briefly in the article about the Ag NPs, which is a not fully bioabsorbable material.

Response: Thank you for your professional advices. We explained that Ag NP were not fully bioabsorbable but biocompatible materials and their metabolic mechanisms in the revised manuscript.

- **Our revision to the manuscript:**

We added “Figure S6a illustrated the degradation and metabolic mechanisms mechanism of BET-suture components.” (see part 2.1 Design and principle of bioabsorbable body-coupling-electrotherapy suture) in the revised manuscript, and added “Mo undergoes oxidative dissolution in physiological fluids to form soluble molybdate ions, which are subsequently excreted via renal clearance. PLGA is hydrolyzed into lactic and glycolic acids, which enter the tricarboxylic acid cycle and are metabolized to CO_2 and H_2O . In contrast, Ag NPs are not fully bioabsorbable; they undergo partial oxidative dissolution to release Ag^+ , which bind to proteins or exist as free ions in plasma and are primarily eliminated through glomerular filtration and urinary excretion.” (see figure caption of Figure S6a) in the revised supplementary materials.

Corresponding changes have been marked in red in the revised manuscript and supplementary materials.

Thank you again for your valuable comments and suggestions.

Response to Reviewer #4

General comment: At its core, this manuscript represents an integration of the authors' previous work published in Science ("Single body-coupled fiber enables chipless textile electronics") and Nature Communications ("A bioabsorbable mechanoelectric fiber as electrical stimulation suture"). However, the integration of these prior studies is not immediately apparent and demands scientific innovation—achievements that are indeed realized in this manuscript. For this reason, these reviewers find the work scientifically compelling and of potential interest to the readership of Nature Communications. Nevertheless, several major concerns must be addressed before the manuscript can be considered for publication.

Response: Thank you for your review and we appreciate your feedback. We have carefully revised the manuscript according to your comments. The replies to each of your concern are listed below.

1. Justification for Use of Ag NPs

There is currently no sufficient justification for the choice of silver nanoparticles. Several questions remain unanswered:

- Why was silver selected over other biocompatible or biodegradable materials, such as Au, Mg, or Zn?
- Why this specific size and shape? The reported diameter range of 10–500 nm is broad. Does this variability impact conductivity, biodegradability, or biocompatibility?
- Were other nanoparticle forms such as nanowires (Ag NWs) or nanostars considered?

The authors are encouraged to provide a rationale grounded in literature or experimental exploration of the influence of nanoparticle size, shape, and composition on system performance, which would significantly strengthen the conclusions.

Response: Thank you for your professional advices. The selection of Ag NPs in this study was based on both literature evidence and experimental comparison, as detailed below:

Material choice: Ag NPs have well-documented biocompatibility, broad-spectrum antibacterial, and anti-inflammatory properties, extensively validated in wound-healing studies (Nature Nanotechnology 2025, 10.1038/s41565-025-01943-y; Nature Biomedical Engineering 2022, 6, 32-43; Nature Communications 2024, 12, 954).

While Au NPs are chemically stable, they lack inherent antibacterial activity and are cost-prohibitive for scalable applications. Zn and Mg NPs possess certain bioactivity but are prone to oxidation and instability, making them unsuitable for improving the dielectric properties of polymer matrices.

In this work, Ag NPs were not employed solely for their antibacterial effects. Instead, their metallic properties were exploited to enhance the dielectric constant of PLGA, enabling

capacitive antibacterial functionality in the BET-suture, while minimizing the high-dose cytotoxicity risks associated with conventional Ag NPs-based antibacterial systems.

Size and shape optimization: The Ag NPs used exhibited a normal size distribution from 10–500 nm, with a mean diameter of ~200 nm (Figure R18a), and their biocompatibility was confirmed (Figure R18b, corresponding to Figure S7 in the original manuscript).

Comparative tests were performed with different Ag NP sizes and Ag nanowires (Ag NWs) at the same loading. PLGA/Ag NPs achieved higher relative permittivity than PLGA/Ag NWs (Figure R18c). Smaller Ag NPs provided greater dielectric enhancement but also exhibited higher cytotoxicity, consistent with literature reports (RECT 2021, 257, 93–119).

At the same concentration (100 ppb), Ag NPs with ~50 nm average size reduced cell viability, whereas particles >100 nm and Ag NWs maintained good biocompatibility (Figure R18d).

Based on these findings, Ag NPs with an average diameter of 200 nm were selected as the optimal trade-off, ensuring: (1) Enhanced dielectric properties for stable ES performance; (2) Sufficient antibacterial functionality; (3) Minimized cytotoxicity risks for long-term wound-healing applications.

Figure R18. (a) The particle size distribution of Ag NPs. (b) Biocompatibility of BET-suture and its components and degradation products. (c) Relative permittivity of BET-suture sheath layers with Ag NPs of different sizes and Ag NWs as nano-doped particles. (d) Live/dead staining images of Ag NPs of different sizes and Ag NWs treating L929 cells for 3 days. Dispersion concentration: 100 ppb, scale bar: 100 μ m. And proportion of live L929 cells after

live/dead staining in each group. $n = 3$ independent samples. All statistical analyzes were performed by one-way ANOVA, data represent mean \pm standard deviation, *** $p < 0.001$, ns indicates not significant.

- **Our revision to the manuscript:**

We added Figure R18a as Figure S11b, Figure R18c and d as Figure S11c and d in the revised supplementary materials. In addition, we added “In addition, we have provided relevant parameters and properties of Ag NPs (Figure S11), and discussed in Note S5 that the effects of different sizes of Ag NPs and silver nanowires on the dielectric properties of the sheath layer, as well as the advantages of selecting Ag NPs.” (see part 2.2 Mechanism of electrical stimulation based on dielectric voltage difference) and “The biocompatibility of BET-suture and Ag NPs of different sizes (50, 200, 500 nm) and Ag NWs (diameter 90 nm, length 2–20 μm) on L929 fibroblasts was also tested with Live/Dead staining.” (see part 4.5 Biocompatibility test) in the revised manuscript, and added “Note S5. Size selection and advantages of Ag NPs” in the revised supplementary materials.

Corresponding changes have been marked in red in the revised manuscript and supplementary materials.

2. Regulation of Maximum Current and Translational Considerations

The manuscript lacks discussion and data on how the system regulates maximum current. While it is understood that the device passively harvests EM energy, in real-world conditions, EM exposure can vary significantly. For example: Hospital environments with strong electromagnetic fields, Industrial settings, Proximity to high-voltage infrastructure (at safe distances). Could such conditions cause excessive current generation, potentially leading to overstimulation or malfunction? Given that this is the third paper from the group on this topic, it is time to begin considering and addressing such translational and safety concerns.

Response: Thank you for raising this important safety concern. We added experimental data on the body-coupled voltage (V_b) and current (I_b) under varying EM field strengths and distances from the source (Figure R19a-c) to illustrate the use safety of BET-suture.

We have supplemented our revised manuscript with experimental data showing the ES current (I_{incision}) at varying distances from an EM source (Figure R19a). As expected, increasing distance leads to greater spatial attenuation of EM waves, resulting in a corresponding decrease in I_{incision} . Importantly, even at minimal distances, the current remains below the microampere level, far lower than the safety threshold for human tissue stimulation, thereby ensuring user safety. This suggests that when exposed to strong EM field environments, maintaining a safe distance will keep the user safe.

Subsequently, we further measured the V_b and I_b of the human body under different EM field strengths while maintaining a distance of 1 m from the EM field. As shown in Figure R19b and c, even when exposed to a strong EM field environment (2.0 kV, 43.6 dBm), the body-coupled voltage is only 15.5 V, and the corresponding current is 16.9 μ A. This further demonstrates that, as long as a sufficient safety distance is maintained, the user safety will be ensured.

In fact, in areas of daily human activity (offices, laboratories, outdoors, factories, etc.), the EM field strength hardly reaches the above levels (2.0 kV, 43.6 dBm). Typically, as shown in Figure R19d (Figure S1b in the original manuscript), the power level of body-coupled environmental EM energy is in the microwatt range (\sim -20 dBm). This power level is insufficient to generate excessive current and does not pose a risk to human health.

We further illustrate that maintaining a safe distance does not pose a hazard to humans with the following formula (Note S3 in the original manuscript):

$$\overrightarrow{V_{incision}} = \overrightarrow{V_b} - \overrightarrow{V_d}$$

At a safe distance, the V_b generated by the body-coupled EM energy will remain within a safe range. The above formula indicates that the entire ES system is powered by body-coupled EM energy, and the V_d generated by the BET-suture is transmitted by V_b to create a potential difference ($V_{incision}$). Therefore, as long as environmental exposure remains within regulated levels (i.e., safe distances are maintained), the induced voltage and current cannot exceed safety limits.

Therefore, both experimental measurements and theoretical considerations confirm that BET-suture can be safely operated in normal living conditions and in high EM field environments where safe distances are maintained.

Figure R19. (a) Peak current (I_p) of ES current (I_{incision}) at different distances from the transmitter. $n = 3$ independent samples. Variation trends of (b) body-coupled voltage (V_b) and (c) current (I_b) under different environmental EM field intensities. (d) Maximum power spectrum of ambient EM waves coupled by the human body and air in various conventional environments, including offices, laboratories, outdoors, factories.

- **Our revision to the manuscript:**

We added Figure R19a as Figure S13j, Figure R19b and c as Figure S2a and b in the revised supplementary materials. In addition, we added “Even under complex E-fields, maintaining a safe distance would ensure personal safety (Figure S2a and b).” (see part 2.1 Design and principle of bioabsorbable body-coupling-electrotherapy suture) and “As the distance increased, the EM wave loss in spatial transmission increased, resulting in a rapid decline in the V_{incision} and I_{incision} (Figure 2g, Figure S13). The voltage and current were at the volt and microampere levels, satisfying ES requirements and ensuring personal safety.” (see part 2.2 Mechanism of electrical stimulation based on dielectric voltage difference) in the revised manuscript.

Corresponding changes have been marked in red in the revised manuscript and supplementary materials.

3. Improvement of Comparison Tables (Table S2 and Table S3)

Table S3 should include quantitative metrics such as wound healing time or the enhancement of healing through electrical stimulation compared to passive controls. Reference data from the authors’ previous studies should be included to facilitate comparison. Additionally:

- Consistent terminology should be used across works. For instance, the therapeutic technique (electrical stimulation) is consistent, even if the delivery method differs.
- Table 2 lacks justification for material selection. Why choose Mo over alternatives like W or Mg? Why PLGA and not Ecoflex or PEG? A clearer rationale supported by quantitative data is needed.

Response: Thank you for your professional advices. We have made the following improvements to Table S3:

Added quantitative healing metrics – We included wound healing acceleration data for different types of functional sutures to facilitate direct comparison. Specifically, antimicrobial sutures were primarily used to reduce inflammation and did not show a clear advantage in healing promotion; passive ES suture showed a 1.58-fold increase in healing speed; and BET sutures showed a 1.92-fold increase in healing speed with no observed infection. This quantitative comparison highlights the superior therapeutic efficacy of our method.

Standardized terminology – The therapeutic technique of electrical stimulation (ES) is now consistently described across the table to avoid ambiguity and enable fair comparison.

Provided clear rationale for material selection – There are two reasons for choosing Mo filament as the electrode: (1) Mo filament has higher tensile strength than W filament and Mg filament (Figure R20a), and its bending stiffness meets the requirements for commercial medical sutures (Figure R20b, corresponding to Figure 1d in the original manuscript); (2) As a bioabsorbable material, Mo is currently the most widely used material after Mg, and it has been extensively reported in various biomedical research fields (Nature 2025, 640, 77-86; Nature Nanotechnology 2021, 16, 708-716; Nature Communications 2024, 15, 1327; Science Advances 2022, 8, eabp9169). Based on the requirements for strength and toughness of sutures and their future clinical potential, Mo filament is more suitable as the electrode for BET-suture.

In addition, PLGA is the most widely adopted clinically approved bioabsorbable polymer, currently used by leading medical device manufacturers (e.g., Johnson & Johnson, Medtronic). By tuning the GA:LA ratio, its degradation rate can be customized to match various wound healing timelines. Compared to alternatives like Ecoflex (non-biodegradable) or PEG (often mechanically weak in dry conditions), PLGA ensures biocompatibility, flexibility, and mechanical stability throughout the healing process. Thus, the selection of PLGA as the sheath material supports both therapeutic functionality and regulatory/clinical translation pathways.

Figure R20. (a) Stress-strain curves of BET-suture prepared using Mo filament, W filament, and Mg filament. (b) Bending stiffness of different types of sutures. n = 3 independent samples.

Table S3. State-of-the-art therapeutic sutures and their properties and mechanisms of action.

Suture materials	Diameter (mm)	Therapeutic techniques	Repair-related signaling pathways	Promote recovery	Bioabsorbable	Ref.
PVA, Ppy, SNAP	0.1-0.4	Drugs or/and growth factors release	Inflammatory cytokines	Reduce infection, without significantly shortening the healing cycle	No	[15]
Silk, QGDs, GM-CSF, CNT	0.22				No	[16]
PGLA, CS, alginate-PAAm	0.54				No	[17]
Tissue fiber, Alg-NHS	0.39				Yes	[3]
PAMPS-AR _x	0.22	Enhanced biocompatibility			No	[18]
PGLAS-NFs-K18	0.38	Photodynamic therapy			No	[4]
Mg, PLGA, PCL	0.35	Passive ES	PI3K/Akt/mTOR MAPK	Healing speed increased by 1.58 times	Yes	[1]
Mo, PLGA, Ag NPs	0.05-0.5	Capacitive antibacterial and controllable ES	Cellular electrophysiology PI3K/Akt/mTOR MAPK Inflammatory cytokines	Healing speed increased by 1.92 times with no obvious infection	Yes	This work

- **Our revision to the manuscript:**

We added Figure R20a as Figure S4a in the revised supplementary materials. In addition, we added the reason for using Mo filament “The electrode, serving as the core layer, plays a decisive role on the strength of BET-suture. Compared to BET-suture prepared using other biodegradable metals (magnesium, Mg and tungsten, W) as electrodes, BET-suture prepared using Mo filament as the electrode exhibited superior tensile strength (Figure S4a).” (see part 2.1 Design and principle of bioabsorbable body-coupling-electrotherapy suture) in the revised manuscript.

Corresponding changes have been marked in red in the revised manuscript and supplementary materials.

4. Clarification of In Vivo Experimental Protocol

The in vivo experiments need a precise description:

- Line 646 indicates rats were euthanized at days 3 and 7, yet the experiment is described as running for 35 days.
- A clear schematic timeline of the experimental protocol should be added.
- Specifically, how were the crocodile clips attached to the animals? Were they secured in a way that guarantees stable and consistent attachment throughout the entire 35-day period?
- What ethical considerations support maintaining such a connection on a live animal for this duration? Is the clip directly attached to the sutures, or another anatomical site? Including detailed images or photographs of the connection method would greatly improve transparency and understanding of the experimental design. If the banana cables are connected directly to the animal's body, this reviewer recommends that the related data be excluded from the manuscript.

Response: Thank you for your professional advices. The purpose of this experiment was to monitor the changes in the ES capacity of BET-suture in vivo over time, which lasted for 35 days until the BET-suture lost its ES capacity. Part of the rats were euthanized on the 3rd and 7th days for inflammation observation and tissue healing assessment, respectively. We added a clear experimental protocol time indication (Figure R21a).

As shown in Figure R21b (Figure S8a in the original manuscript), we left one end of the BET suture outside the body in three rats from the same batch that had been sutured, for long-term monitoring of ES performance. As shown in Figure R21c (Figure S27b in the original manuscript), we temporarily anaesthetized the rats on days 1, 3, 7, 11, 14, 17, 21, 28, and 35, during which time we temporarily attached alligator clips to the externalized end of the suture. The clips were never connected directly to the animal's skin or body tissue. After awakening, we used an oscilloscope to read the value of V_d and calculate $V_{incision}$ (Figure R21d, corresponding to Figure S27a in the original manuscript). This method minimized discomfort

and prevented inconsistent contact. Except during the testing period (within 15 minutes), all rats were housed uniformly in an accredited animal facility at Shanghai Sixth People's Hospital.

We fully recognize the ethical importance of minimizing animal distress. The following steps were taken: (1) No clips were permanently attached to any part of the animal's body. (2) Monitoring was performed under short-term anesthesia only during testing days. (3) Animals were housed in a licensed and accredited animal facility (Shanghai Sixth People's Hospital) under standard conditions. (4) All animal experiments were performed according to protocols approved by approved by Committee on Ethics of Donghua University (DHUEC-NSFC-2022-43).

Figure R21. (a) The schematic timeline of the animal experiment protocol. (b) Optical image of BET-suture stitched to the subcutaneous muscle of SD rats (red dashed box). One end of the BET-suture was left outside the body after stitching and was used to connect to an oscilloscope for real-time measurement of electrical signals. (c) Variation of V_p at different times of implantation. $n = 3$ independent samples. (d) Photos of $V_{incision}$ measured in real time after 1, 10 and 28 days of BET-suture implantation.

- **Our revision to the manuscript:**

We added Figure R21a as Figure S26a in the revised supplementary materials. In addition, we added “The specific schematic timeline of the experimental protocol is shown in Figure S26a.” (see part 2.5 Evaluation of incision healing effect in vivo) in the revised manuscript.

Corresponding changes have been marked in red in the revised manuscript and supplementary materials.

5. Terminology Inconsistency (ES-Suture vs BET-Suture)

The term "ES-suture" is confusing, as it was previously used to describe the mechanically activated suture. In this manuscript, it appears to refer to a BET-suture without AgNPs. Consistent and clearly defined terminology is essential to avoid confusion.

Response: Thank you for your careful review. We have replaced “ES-suture” with “BS-suture” in the naming of the animal experiments to avoid confusion with the previous mechanically activated suture.

Corresponding changes have been marked in red in the revised manuscript and supplementary materials.

6. Definition of "Optimal Recovery Status" (Figure 5b, Line 413–414)

The claim that the BET-suture shows “optimal recovery status” needs a more precise and evidence-based explanation. What metrics define "optimal"? Histology? Functional recovery? Please elaborate clearly.

Response: Thank you for your professional advices. In Figure R22a (Figure 5b in the original manuscript), the optimal recovery state refers to the original conclusion of tissue healing based on the appearance of the tissue after 7 days of surgery. The BET-suture group demonstrated the optimal recovery status, characterized by tissue color closest to normal, with no signs of redness, swelling, exudation, or necrosis. Healing was tight, with a smooth, continuous tissue surface, flat adhesion, and an appearance nearly identical to normal tissue, significantly superior to the Control, AB-suture, and BS-suture groups.

Similarly, H&E and Masson staining results also showed that the wound stitched with BET suture had lower inflammatory infiltration and better collagen deposition and alignment (Figure 22b, corresponding to Figure 5c in the original manuscript). These results confirm that the BET-suture not only supports rapid wound closure but also promotes a favorable internal tissue remodeling environment, justifying the use of the term “optimal recovery.”

Figure R22. (a) Optical photographs of the incisions stitched in each group and the incisions and their surrounding muscle tissues removed in each group after 7 days of stitching. Scale bar: 2 cm. (b) H&E and Masson staining images of the incision and its surrounding tissues in each group. Scale bar: 500 μm .

- **Our revision to the manuscript:**

We added the explanation “A detailed comparative analysis of the recovery status and anti-inflammatory properties of the three different suture materials is provided in Note S6.” (see part 2.5 Evaluation of incision healing effect in vivo) in the revised manuscript, and added “The BET-suture group demonstrated the optimal recovery status, characterized by tissue color closest to normal, with no signs of redness, swelling, exudation, or necrosis. Healing was tight, with a smooth, continuous tissue surface, flat adhesion, and an appearance nearly identical to normal tissue, significantly superior to the Control, AB-suture, and BS-suture groups.” of the “optimal recovery status” to Note S6 in the revised supplementary materials.

Corresponding changes have been marked in red in the revised manuscript and supplementary materials.

7. Sample Clarification (Figure S25)

The statement that $n = 3$ independent samples were used for mouse measurements requires clarification. Does this mean three different animals, or three repeated measurements from a

single animal? The sample size and method of measurement should be clearly stated.

Response: Thank you for your careful review. In Figure S25, $n = 3$ independent samples refer to the average results of electrical signal measurements in three different rats. In addition, all independent samples in this study represent the number of samples, tissues or animals in each group, with only one result obtained for each sample or animal.

- **Our revision to the manuscript:**

We added the explanation “All independent samples in this study represent the number of samples, tissues or animals in each group, with only one result obtained for each sample or animal.” (see part 4.14 Statistical analysis) of the samples in the revised manuscript.

Corresponding changes have been marked in red in the revised manuscript.

Minor Comments

8. The novelty of this work compared to the previous two papers is not clearly articulated. A concise summary should be added to the Introduction, explicitly stating: What was achieved in each prior paper, what limitations remained, How this work addresses those limitations. Specifically, the key limitation from the Science paper—insufficient current from EM energy harvesting—seems to be overcome in this work via Ag NP-enhanced conductivity. If this is indeed the core novelty, it should be clearly emphasized in the abstract and intro.

Response: Thank you for your professional advices. We have explained the achievements and limitations of the previous two papers in the introduction. The self-powered ES sutures developed by our group (Nature Communications 2024, 15, 8462) to modulate cell proliferation and migration have overcome the chemical hazards associated with antimicrobial sutures and the need for an external power source. However, since the mechanoelectric system relied on external mechanical stimulation to generate energy, it could only be applied in dynamic or specific activation scenarios.

Subsequently, our group has proposed a novel form of energy interaction of body-coupled fiber (Science 2024, 384, 74-81). Inexhaustible electromagnetic (EM) energy could be harvested and transmitted wirelessly through the fiber-body interface. However, all such energy interaction modes were realized through external media between the human body and the environment, without delving into the interaction form and application value of implantable electronic devices and electromagnetic energy (Figure R1).

In this work, we explored the mechanism of EM energy coupling within the body using implantable electronic fiber and achieved the ability to suture and treat wounds in vivo. This has opened up a new direction for body-coupled EM energy sources in the medical field. This entirely physical therapy addresses the issues of chemical antimicrobials and the overuse of

silver nanoparticles as antimicrobial agents, represents a significant advance in surgical practice.

- **Our revision to the manuscript:**

We added “We have enhanced the dielectric properties of the BET-suture using safe amounts of silver nanoparticles (Ag NPs), maximizing its ES and antimicrobial capabilities.” to the introduction in the revised manuscript.

Corresponding changes have been marked in red in the revised manuscript.

9. The introduction of Ag NPs in the Results section is abrupt. Given their biocompatibility profile, they should be introduced earlier. Since the data supports their safe use in this context, this should be highlighted as a novel achievement in both the abstract and intro.

Response: Thank you for your professional advices. We have highlighted that the therapeutic approach of this work addresses the safety of Ag NP use in the abstract and introduction.

- **Our revision to the manuscript:**

We added “This entirely physical therapy addresses the issue of excessive use of silver nanoparticles.” to the abstract and “This physical antimicrobial strategy addresses the potential biological hazards associated with the overuse of Ag NPs as antimicrobial agents.” to the introduction in the revised manuscript.

Corresponding changes have been marked in red in the revised manuscript.

10. Figure S2 is currently schematic. The authors are encouraged to include actual images or short videos of the fabrication process. This will help the reader better understand the reproducibility and reliability of the method.

Response: Thank you for your professional advices. We have added actual images of the preparation process. Figures R23a and b showed the physical photos of the preparation process of primary fibers and BET-suture, respectively. We used the constructed wet-coating equipment to coat PLGA/Ag NPs onto Mo filaments, and then twisted a certain number of primary fibers using a doubling and twisting combined testing machine (Anytester AT208, China) to obtain BET-suture.

Figure R23. (a) Physical illustration of the preparation of primary fiber (Mo@PLGA/Ag NPs) using the constructed wet-coating device. (b) Process of preparing BET-suture by twisting the primary fibers using a doubling and twisting combined testing machine.

● **Our revision to the manuscript:**

We added Figure R23a and b as Figure S3b and c in the revised supplementary materials. In addition, we added the equipment model “using a doubling and twisting combined testing machine (Anytester AT208, China)” (see part 4.2 Fabrication of BET-suture) in the revised manuscript.

Corresponding changes have been marked in red in the revised manuscript and supplementary materials.

11. (Line 65) The sentence: “But scenarios that require dynamic conditional use limit its application to restricted area,” is vague. What are these scenarios? Please provide concrete examples and references, or revise the sentence for clarity.

Response: Thank you for your professional advices. We have rewritten this sentence “But scenarios that require dynamic conditional use limit its application to restricted area,” as “However, since the mechanoelectric system relied on external mechanical stimulation to generate energy, it could only be applied in dynamic or specific activation scenarios.” (see part Introduction) to more clearly express the limited application range of the mechanoelectric system.

Corresponding changes have been marked in red in the revised manuscript.

12. Figure 1b: Scale bar is not visible.

Response: Thank you for your careful review. We have changed the color of the scale bar in Figure 1b to black for clearer presentation.

13. Figure 2a: The conductive path and Ag NPs orientation are not clearly illustrated.

Response: Thank you for your careful review. Figure 2a showed that Ag NPs were uniformly

distributed inside the sheath layer. Excessive addition of Ag NPs would form certain conductive paths, causing excessive leakage current and reducing the electrical stimulation performance of BET-suture. We explained this in the figure caption of Figure 2a.

- **Our revision to the manuscript:**

We added “Appropriate Ag NPs enhance the dielectric properties, while excessive addition of Ag NPs will form certain conductive paths.” to the figure caption of Figure 2a in the revised manuscript.

Corresponding changes have been marked in red in the revised manuscript.

14. Figures 3 & 4: Staining images lack visibility and should be enhanced for clarity.

Response: Thank you for your careful review. We have further extracted higher resolution versions of Figure 3 and Figure 4 and provided them in the revised manuscript.

15. Figure 5a: A clearer image of the mouse, including visualization of suture placement and a timeline of the experimental procedure, would be helpful as discussed.

Response: Thank you for your professional advices. We have provided the clearer image of the mouse, including visualization of suture placement and a timeline of the experimental procedure in comment 4.

16. Consider including a comparative timeline in Figure 1 or visual summary of suture technologies—from antimicrobial chemical sutures to the current work with physical/electrical stimulation. This would help multidisciplinary readers position this work in the broader technological landscape, including existing limitations and the pathway to clinical translation.

Response: Thank you for your professional advice. We have added a visual summary of suture techniques in Figure 1, from antimicrobial chemical sutures to self-powered ES sutures to the current body-coupling-electrotherapy sutures, and explained the relevant advantages and existing limitations.

Figure R24. The development of suture technology—from antimicrobial chemical suture (Anti-B) technology to self-powered ES (BioES) suture to the currently used body-coupling-electrotherapy (BET) suture.

- **Our revision to the manuscript:**

We added Figure R24 as Figure 1e in the revised manuscript. In addition, we added “Compared to previous antimicrobial sutures and self-powered ES sutures, the BET-suture introduces a novel, comprehensive electrotherapy mode for wound healing management, adaptable to various therapeutic scenarios under both static and dynamic conditions, making it better aligned with clinical practice (Figure 1e).” (see part 2.1 Design and principle of bioabsorbable body-coupling-electrotherapy suture) in the in the revised manuscript.

Corresponding changes have been marked in red in the revised manuscript.

Thank you again for your valuable comments and suggestions.

Response to Reviewer #5

General comment: I co-reviewed this manuscript with one of the reviewers who provided the listed reports. This is part of the Nature Communications initiative to facilitate training in peer review and to provide appropriate recognition for Early Career Researchers who co-review manuscripts.

Response: Thank you for your joint review. We have carefully revised the manuscript based on the comments of the above three reviewers. Each of their concerns has been responded to one by one.

Response to Reviewer #6

General comment: I co-reviewed this manuscript with one of the reviewers who provided the listed reports. This is part of the Nature Communications initiative to facilitate training in peer review and to provide appropriate recognition for Early Career Researchers who co-review manuscripts.

Response: Thank you for your joint review. We have carefully revised the manuscript based on the comments of the above three reviewers. Each of their concerns has been responded to one by one.

Responses to the Reviewers' Comments

We sincerely thank the reviewers for their careful and thorough review. We have revised our manuscript very carefully in the light of their suggestions and comments.

The following responses have been prepared to address all of the reviewers' comments in a point-by-point fashion. (Comments in black, responses in blue):

Response to Reviewer #1

General comment: The authors have revised the manuscript in response to the reviewers' comments. However, several important issues remain to be addressed:

Response: Thank you for your review. We have systematically replied to the important issues that you concern below.

1. Body-coupled or body-mediated electronic systems are already well-established technologies. Similarly, PLGA and Mo are widely used bioresorbable materials. The authors should clearly articulate the novelty of their work beyond these existing approaches.

Response: In our previous response, we have already elaborated in detail on the innovations of BET-suture in the fields of bioabsorbable implantable devices and body-coupled systems in terms of materials and mechanisms.

Beyond the aforementioned innovations, BET-suture represents a major breakthrough in the field of smart medical suturing.

By 2024, innovation in surgical sutures has primarily focused on the chemical antimicrobial sutures. We have already highlighted the hazards of this type of suture in the introduction "The utilization of chemical antimicrobial sutures, encompassing antibiotics²⁰, metal antimicrobials²¹, photodynamic therapy (PDT)^{22,23}, has been widely reported as a means to modulate immune responses. However, persistent chemical hazards, such as tissue drug resistance, cytotoxicity of excessive metal nanoparticles and PDT molecules, are hindering the development of effective antimicrobial sutures.". Subsequently, to address the challenges associated with these chemical antimicrobial sutures, we presented a novel mechanoelectric system-based ES surgical suture that garnered significant industry recognition (Figure R1) (Nature Communications 2024, 15, 8462). Our proposed concept of electrotherapy-enhanced healing represents a major innovation in the field of smart sutures. However, previous work still faced challenges in achieving full-stage healing regulation and encountered limitations in application scenarios. We have elaborated on this in the introduction "In response to these challenges, the self-powered ES sutures developed by our group¹⁹ to modulate cell proliferation

and migration have overcome the chemical hazards associated with antimicrobial sutures and the need for an external power source. However, since the mechanoelectric system relied on external mechanical stimulation to generate energy, it could only be applied in dynamic or specific activation scenarios.”.

In this work, BET-suture introduces several key innovations:

1. Beyond chemical antimicrobials: Previous advances in sutures have mainly relied on chemical strategies (antibiotics, metallic ions, PDT), which suffer from cytotoxicity and drug resistance. BET-suture instead achieves antimicrobial activity via capacitive charge mediation, avoiding persistent chemical hazards.

2. Beyond mechanoelectric sutures: Our earlier self-powered ES sutures required external mechanical activation and were limited to dynamic scenarios. BET-suture, for the first time, leverages body-coupled energy conversion to provide continuous and scenario-independent stimulation *in vivo*.

3. Comprehensive wound regulation: By combining a conductive core with a high-dielectric sheath, BET-suture establishes a stable potential difference that enables both electrotherapy-enhanced healing and capacitive antibacterial protection. This dual functionality supports full-stage wound regulation (from inflammation to remodeling), which has not been reported in prior intelligent suture studies (Table S3).

As summarized in Figure R1b (corresponding to Figure 1e in the first revised manuscript), BET-suture marks a breakthrough in the developmental trajectory of suture technology by offering a safe, universal, and versatile platform for smart surgical wound management.

Figure R1. (a) Interviews and coverage by major renowned international media outlets. (b)

The development of suture technology—from antimicrobial chemical suture (Anti-B) technology to self-powered ES (BioES) suture to the currently used body-coupling-electrotherapy (BET) suture.

2. In Figure R2, the reported peak voltage exceeds 15 V, which is unusually high for an implantable system. In particular, when an individual passes near high-voltage power lines (typically above 230 kV), there may be potential safety hazards for the user. The authors should address this concern.

Response: Thank you for your concern regarding safety. The reported 15 V in Figure R2 reflects the maximum body-coupled voltage under amplified laboratory conditions (EM field of 2 kV, 43 dBm, at 1 m). In practical environments, the induced voltage decreases rapidly with weaker EM fields or increased distance (Figure R2a and b), and the stimulation intensity of BET-suture can be adjusted to meet therapeutic requirements, ensuring clinical safety.

Regarding exposure near high-voltage power lines (e.g., 230 kV), such voltages indeed represent hazardous transmission conditions (Figure R2c). However, according to the quasi-static induction model:

$$E_b = \frac{V}{h \cdot \ln\left(\frac{2h}{r}\right)}$$

where V is the line-to-ground voltage, h is the distance from the conductor, and r is the cable radius.

Based on IEC 60840-2023 ($r \approx 15$ mm for 230 kV) and ICNIRP guidelines (safe limit: 5 kV/m at 50 Hz) (Health Physics 2010, 99, 818-836), the minimum safety distance is ~ 6.8 m. National standards (DL/T 5092-1999) mandate conductor-to-ground clearances exceeding this threshold (Table R1), meaning areas accessible to humans inherently satisfy safety limits.

Therefore, BET-suture harvests EM energy only within environments already deemed safe for human exposure by international and national regulations, and does not pose additional safety risks during implantation or clinical use.

Figure R2. Variation trends of body-coupled voltage (V_b) with changes in strength of the environmental EM field. (b) V_p of $V_{incision}$ at different distances from the transmitter. (c)

Minimum distance between the human body and high-voltage cables.

Table R1. Voltage levels and minimum conductor-to-ground clearances.

Voltage levels	Minimum conductor-to-ground clearances
35~110 kV	~7 m
220 kV	~7.5 m
330 kV	~8.5 m
500 kV	~14 m
750 kV	~19.5 m
1000 kV	~70-80 m

3. In Figure R3b, it is not clear why the PLGA/Ag NP structure does not outperform the Mo-based structure. Further clarification and mechanistic explanation are required.

Response: As shown in Figure R3a (corresponding to Figure 3b in the first revised manuscript), the combined device configuration was used to measure transient charge transfer density and charge accumulation in the sheath layer. For standardization, Mo was placed between copper sheets as a pure conductive reference (gray dashed line in Figure R3b) to quantify the signal generator output.

Accordingly, the PLGA/Ag NPs sheath layer is not expected to “outperform” Mo, since Mo functions solely as a control conductor rather than as a dielectric material for comparison. The key comparison lies among sheath materials, where the incorporation of Ag NPs significantly enhances transient charge transfer density and charge accumulation (Figures R3b and R3c, corresponding to Figure S17d and Figure 3c in the first revised manuscript). This improved capacitive response is expected to support effective antibacterial performance of BET-suture.

Figure R3. (a) Diagram of experimental setup for measuring the charge storage capacity of the dielectric layer. (b) Real-time transferred charge densities of Mo, PLGA and PLGA/Ag NPs at a fixed AC. (c) The relationship between the stored charges and time. Effective voltage $5 V_{rms}$, frequency 50 Hz.

- **Our revision to the manuscript:**

We explain the role of Mo “Mo serves as the control reference (gray dashed line) to quantify the transient charge transfer density from the signal generator.” in the figure caption for Figure S17d.

Corresponding changes have been marked in red in the revised manuscript and supplementary materials.

4. The claim that the charge storage and release capacity of Ag NPs contributes to antibacterial properties is not convincing. Since electrical stimulation alone does not effectively reduce bacterial populations, the antibacterial effect should be evaluated under conditions without electrostimulation using the BET. If antibacterial activity is observed, it is more likely attributable to the intrinsic toxicity of Ag NPs rather than their charge-storage capability.

Response: As addressed in our antimicrobial experiments, we tested the unenergized BET-suture with PLGA/Ag NP sheath as a control (Figure R4, corresponding to Figure S18 in the first revised manuscript). Part 2.3 describes that the unenergized BET-suture does not achieve bactericidal efficacy “S. aureus and E. coli exhibited robust growth in the control group, revealing that the Ag NPs embedded within the PLGA were unable to reach the bacteria and thus demonstrated suboptimal antimicrobial properties.”.

In contrast, the energized BET-suture demonstrated significant antibacterial effects. This

comparison confirms that the antimicrobial performance originates from the enhanced dielectric properties of Ag NPs-containing sheaths, which enable capacitive charge-mediated antibacterial activity, rather than from the inherent toxicity of Ag NPs.

Perhaps the naming of the “control” group caused confusion in your review, so we have revised it in the latest revised manuscript.

Figure R4. Capacitive antimicrobial capacity of BET-suture. (a) Photographs of the proliferative state of *S. aureus* and *E. coli* on LB plates after treatment in each group. Scale bar: 1 cm. Cell counting statistics for (b) *S. aureus* and (c) *E. coli* on LB plates. $n = 3$ independent samples. All statistical analyses were performed by one-way ANOVA, data represent mean \pm standard deviation, ** $p < 0.01$, * $p < 0.05$, ns indicates not significant.

● **Our revision to the manuscript:**

We replaced “Control” with “BET unenergized” in Figures 3e-g, Figure S18, Figure S19, and the corresponding descriptions “To validate the capacitive antibacterial properties of BET-suture, bacteria were treated for 3 h using energized BET-suture (BET group), energized BET-suture without Ag NPs (BET no-Ag NPs group), and unenergized BET-suture (BET unenergized group).” (see part 2.3 Charge-enhanced antimicrobial effect based on capacitive properties).

Corresponding changes have been marked in red in the revised manuscript and supplementary materials.

5. Figure R5b is confusing. In Figure 5, the applied peak voltage of 5 V corresponds to approximately 9 V/mm of electric field (see Figure R5a). Did the authors calculate the electric field under the actual animal experimental conditions? Figure R5b suggests that an electric field exceeding 5 V/mm can cause cellular damage. Therefore, the safety of the system should be carefully analyzed and discussed.

Response: In our previous response, we clarified that the E-field threshold was determined through simulations and cell experiments, which identified an effective range of 0.75–5 V/mm for promoting proliferation and migration. We also performed corresponding simulations for animal conditions.

In part 2.2, we have systematically measured the electrical signal output and simulated and statistically analyzed the electrical stimulation intensity in SD rats. As shown in Figures R5a-c (corresponding to Figures 2f, g and Figure S14a in the first revised manuscript), the applied electrical stimulation intensity in animal experiments was derived from these simulation and statistical results. The incision size used in this simulation and statistical analysis corresponds to the incision size described in part 4.12 Animal experiments “The skin was slashed with a scalpel and an incision of 3 cm long and 1 cm deep was made vertically in the muscle.”. These results guided the stimulation parameters used in animal studies.

Therefore, as shown in Figure R5d (Figure 5d in the first revised manuscript), based on simulation results, we employ a stimulation voltage of approximately 5 V (distance: about 35 cm). The resulting electric field strength is around 2 V/mm. This value lies well within the effective therapeutic range (0.75–5 V/mm) and below the level associated with cellular damage (>5 V/mm), thereby confirming both efficacy and safety of the BET-suture system in vivo.

In addition, we have also discussed the use of this system and the safe stimulation range in part 2.2 “The average E-field strength generated by the simulation at different distances was calculated and counted (Figure S14a). The results demonstrated that beyond 60 cm, the average E-field strength was below the threshold for effective ES (>0.75 V/mm, as measured in part 2.4). By adjusting the distance appropriately, the ES intensity can be maintained within the effective range.”.

Figure R5. (a) Finite element simulation of E-field strength generated by BET-suture at incision site. The spiral line represents BET-suture and the rectangle represents incision. (b) V_p of $V_{incision}$ at different distances from the transmitter. (c) The maximum, minimum and median values were taken from the E-field results of the COMSOL simulation and the average E-field strength was calculated. (d) Real-time signals of $V_{incision}$ continuously monitored over 7 days of stitching.

Thank you again for your valuable comments and suggestions.